# Asymmetric nucleosome PARylation at DNA breaks mediates directional nucleosome sliding by ALC1

Luka Bacic[1], Guillaume Gaullier [1,3], Jugal Mohapatra[2], Guanzhong Mao[1], Klaus Brackmann[1], Mikhail Panfilov [1], Glen Liszczak[2], Anton Sabantsev [1] ✉ & Sebastian Deindl [1] ✉

The chromatin remodeler ALC1 is activated by DNA damage-induced poly(ADP-ribose) deposited by PARP1/PARP2 and their co-factor HPF1. ALC1 has emerged as a cancer drug target, but how it is recruited to ADP-ribosylated nucleosomes to affect their positioning near DNA breaks is unknown. Here we find that PARP1/HPF1 preferentially initiates ADP-ribosylation on the histone H2B tail closest to the DNA break. To dissect the consequences of such asymmetry, we generate nucleosomes with a defined ADP-ribosylated H2B tail on one side only. The cryo-electron microscopy structure of ALC1 bound to such an asymmetric nucleosome indicates preferential engagement on one side. Using single-molecule FRET, we demonstrate that this asymmetric recruitment gives rise to directed sliding away from the DNA linker closest to the ADP-ribosylation site. Our data suggest a mechanism by which ALC1 slides nucleosomes away from a DNA break to render it more accessible to repair factors.

The pronounced cytotoxicity of DNA double-strand breaks represents a great threat to genome integrity and can rapidly overwhelm the cellular DNA repair capacity. A key requirement for the successful repair of double-strand breaks is their rapid recognition and adequate cellular signaling, in the absence of which the cell triggers cell death or apoptosis[1]. Among the important players in the early response to double-strand breaks are the ADP-ribosyltransferases PARP1 and PARP2. These enzymes sense DNA breaks and signal their presence by attaching poly(ADP-ribose) (PAR) onto themselves and their target proteins, including histones[2–4]. PARP enzymes have shown great promise as therapeutic targets of anticancer therapy. Despite the notable clinical success of PARP inhibitors, homologous-recombination-deficient (HRD) cancer cells can still develop resistance[5].

The early response to DNA lesions typically involves PAR- and ATP-dependent chromatin remodeling[6,7]. The chromatin remodeler (remodeler) ALC1 (Amplified in Liver Cancer 1) has recently emerged as a key player required at DNA damage sites. ALC1 exerts its catalytic activity through a conserved Snf2 (sucrose non-fermenter 2)-like ATPase domain and binds to PAR chains near DNA lesions with its macro domain[8,9]. In the absence of PAR, the macro domain of ALC1 is placed against its ATPase, stabilizing an inactive conformation[10,11]. Structural, biochemical and in vivo scrutiny suggested that the macro domain of ALC1 interacts with its C-terminal ATPase lobe[10,12]. Upon recruitment to DNA damage sites, the macro domain of ALC1 binds to PAR[8,9], which displaces the macro domain from the ATPase domain and releases ALC1 autoinhibition[10,11]. Full activation of ALC1 requires the insertion of an Arginine anchor from the ALC1 linker into the acidic patch of the nucleosome[12–14]. A PARylation response triggering efficient repair of DNA damage requires HPF1 (Histone PARylation Factor 1)[15]. The crystal structure of PARP2 and HPF1 indicates that HPF1 completes the PARP2 active site to redirect ADP-ribosylation toward serine in the KS motif[15–17]. Serine ADP-ribosylated sites were also found in the histone H3 and H2B tails of nucleosomes[18]. Nucleosomal histone PARylation in the presence of HPF1 is therefore most likely a

[1]Department of Cell and Molecular Biology, Science for Life Laboratory, Uppsala University, 75124 Uppsala, Sweden. [2]Department of Biochemistry, The University of Texas Southwestern Medical Center, 5323 Harry Hines Boulevard, Dallas, TX 75390, USA. [3]Present address: Department of Chemistry - Ångström, Uppsala University, 75120 Uppsala, Sweden. ✉e-mail: anton.sabantcev@icm.uu.se; sebastian.deindl@icm.uu.se

prerequisite for the activation of ALC1 as a crucial step in the DNA damage response[13].

ALC1 is emerging as an important target for therapeutic intervention strategies in cancer, since ALC1 inactivation exacerbates the cytotoxic effects of clinical PARP inhibitors in HRD cancer cells[19–23]. There is therefore considerable interest in the molecular analysis of ALC1 and its targeting by structure-based drug design. Recent cryo-electron microscopy (cryo-EM) studies of ALC1 bound to an ADP-ribosylated nucleosome, enzymatically modified in vitro using PARP2 and HPF1, enabled the visualization of several intermediate states of the ALC1 ATPase motor from the recognition of the ADP-ribosylated nucleosome to the tight binding and activation[13].

However, little is known about how ALC1 remodeling affects nucleosome positioning near a DNA break. ALC1 plays a role in chromatin relaxation at sites of DNA damage[7,24,25], which is thought to promote repair[26–29]. In order to facilitate access to the repair machinery, ALC1 may slide nucleosomes away from DNA breaks[13]. Importantly, a direct demonstration of such a preferential directionality is lacking, and the underlying mechanism is completely unknown.

Here, we leveraged the homogenous site- and degree-specific ADP-ribosylation of recombinant histones[30] to examine ALC1-induced nucleosome sliding at DNA breaks. We observed that PARP1/HPF1 preferentially ADP-ribosylates the break-proximal H2B tail, which asymmetrically recruits ALC1 to one side to slide the nucleosome away from the break.

## Results

### ADP-ribosylation by PARP1/HPF1 is more readily initiated on the H2B proximal to the nearest DNA end

In order to probe PARP1/HPF1 activity on either side of a nucleosome next to a free DNA end mimicking a double-strand break, we generated asymmetric nucleosomes with distinct H2A/H2B dimers[31,32] (Fig. 1). We first assembled oriented hexasomes using an H2A/H2B dimer with a Cy3-labeled H2B. The single Cy3-H2A/H2B dimer was homogeneously inserted on the more bendable (strong) side of the asymmetric 601 positioning sequence[31–34], the side with the shorter DNA linker (3 bp) mimicking a double-strand break. We reconstituted nucleosomes by adding a dimer containing Cy5-labeled H2B, which was incorporated on the longer-linker (78 bp) side (Fig. 1a). Following the addition of PARP1/HPF1 and limiting concentrations of NAD$^+$, we separated the

proteins by SDS-PAGE and detected the two copies of H2B based on their distinct fluorescence (Fig. 1b). ADP-ribosylation of H2B decreased its mobility, allowing us to quantify the extent of histone ADP-ribosylation on both the shorter- (Cy3 signal) and longer-linker (Cy5 signal) sides of the nucleosome. Interestingly, ADP-ribose chains were more efficiently elongated on the longer-linker side of the nucleosome (Fig. 1b). More strikingly, the extent of ADP-ribosylation initiation, as judged by the disappearance of non-ADP-ribosylated H2B, was higher proximal to the shorter-linker side at all NAD$^+$ concentrations (Fig. 1c). At the highest NAD$^+$ concentration (32 μM), essentially all short linker-proximal histone H2B was modified, while about 40% of the long linker-proximal H2B remained unmodified. The concentration of HPF1 in the cell is 20-fold lower than that of PARP1, and HPF1 is required for initiation, but not elongation of histone PARylation[35]. Initiation therefore likely represents the rate-limiting step for histone PARylation in vivo. Thus, the observed preference for initiating PARylation on the double-strand break-proximal H2B tail is expected to result in the installation of ADP-ribose chains mostly on one side of the nucleosome.

### Cryo-EM reveals ALC1 bound on one side of the asymmetrically ADP-ribosylated nucleosome

Could the observed asymmetry in histone H2B PARylation result in preferential recruitment of ALC1 to one side of the nucleosome? To address this question, we sought to determine the structure of ALC1 bound to a nucleosome with a homogeneously ADP-ribosylated H2A/H2B dimer on one side. We combined oriented hexasomes featuring an unmodified H2A/H2B dimer on the strong side (Fig. 2a) with an H2A/H2B dimer modified with a tri-ADP-ribose chain attached to H2B Ser6. We incubated the resulting nucleosomes with ALC1 and ADP-BeF$_x$ and subjected the sample to cryo-EM structure determination.

The cryo-EM map revealed an ALC1 ATPase tightly engaging the nucleosomal DNA at superhelical location (SHL) 2, where the SWI/SNF, ISWI, and Chd1-type remodelers all translocate DNA[36–40] (Fig. 2b). The map also showed an interaction between the C-terminal ATPase lobe and the N-terminal H4 tail. Both of these features are conserved among virtually all structurally characterized remodelers[41–50].

Our map, with an overall resolution of 3.0 Å, is substantially improved compared to our previously published cryo-EM analyses of ALC1 bound to an enzymatically ADP-ribosylated nucleosome[13]. The map indicates that only a single ATPase is bound to the nucleosome. Importantly, the local resolution in the DNA region is sufficient to distinguish purines from pyrimidines (Supplementary Fig. 2b), which enabled us to unambiguously determine that ALC1 is bound to the DNA at the strong side, close to the tail of the ADP-ribosylated H2B at the weak side (Fig. 2c). In addition, quantifying the occupancy of this map's features with the recently published algorithm OccuPy[51] revealed that the acidic patch on the entry side contains additional density, while the acidic patch on the other side does not (Supplementary Fig. 3). This is consistent with our previous biochemical data, which showed that an interaction between the linker of ALC1 and the entry-side acidic patch is required for ALC1-catalyzed nucleosome sliding[14]. Our cryo-EM map does not allow us to locate the macro domain, presumably due to the fact that both the H2B N-terminal tail and the linker of ALC1 are too long and flexible to constrain the macro domain to a well-defined location. Nonetheless, based on the design of the asymmetrically, site-specifically and homogeneously ADP-ribosylated nucleosome and the low-nanomolar affinity of the macro domain for tri-ADPr[11], the macro domain must be bound to the H2B ADP-ribosylated site on the weak side of the 601 sequence. In fact, the observed location of the ATPase places its C-terminus closer to the N-terminal H2B tail on the weak side (Fig. 2c). Notably, there was no residual density on the other SHL2 site, contrary to what we observed previously with a heterogeneously ADP-ribosylated nucleosome[13]. The site where the macro domain binds oligo-ADPr therefore appears to

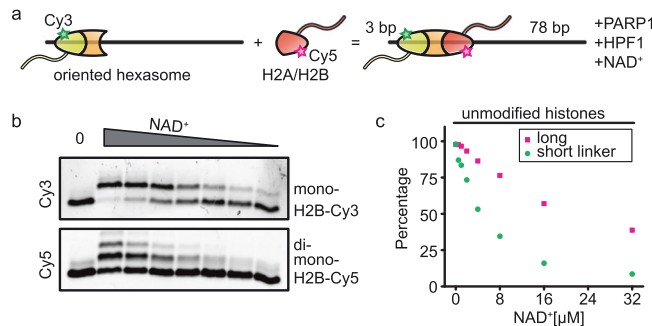

**Fig. 1 | PARP1/HPF1 initially installs the ADP-ribose chain predominantly on the short linker-proximal H2B tail. a** Schematic of nucleosome assembly *via* the oriented incorporation of histone dimer. **b** SDS-PAGE gel imaged by Cy3 and Cy5 fluorescence detection, resolving the histones after PARylation by PARP1/HPF1 at different NAD$^+$ concentrations (0.5, 1, 2, 4, 8, 16, and 32 μM). One representative out of two independent experiments is shown (see also Supplementary Fig. 1). **c** Percentage of remaining non-ADP-ribosylated H2B as a function of NAD$^+$ concentration on the longer-linker (magenta squares) and shorter-linker (green circles) side. Percentages represent the fraction of non-ADP-ribosylated H2B, calculated as the integrated intensity of the band from non-ADP-ribosylated H2B divided by the total integrated intensity from all H2B bands (ADP-ribosylated and non-ADP-ribosylated). Source data are provided as a Source Data file.

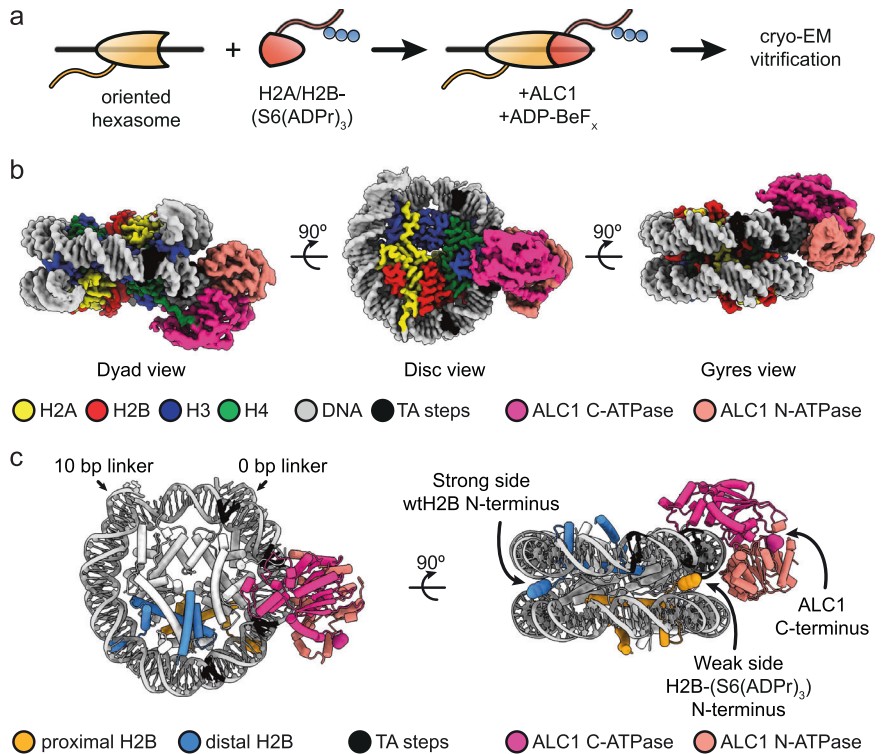

**Fig. 2 | ALC1 engages the SHL 2 location proximal to the oligo-ADP-ribosylated H2B tail. a** Cryo-EM sample preparation. **b** Cryo-EM map of the complex between a uniquely ADP-ribosylated nucleosome and ALC1, shown at a contour level of 0.05 and colored by chain assignment (H3: blue, H4: green, H2A: yellow, H2B: red; DNA: gray, ALC1: pink). Black: TA steps of the strong side of the 601 sequence. **c** Atomic model shown in disc and gyres views (distal H2B: blue, proximal H2B: orange, and ALC1: pink). The N-terminal residue of the two copies of H2B and the C-terminal residue of ALC1 are shown as spheres and indicated by arrows.

determine which of the two SHL2 sites the ATPase engages, and which of the two acidic patches the Arginine anchor probes.

**The location of an oligo-ADP-ribose chain relative to the nucleosome core can bias the remodeling directionality of ALC1**

Given its role in the DNA damage response, we reasoned that ALC1 is likely to have a preference for sliding nucleosomes away from DNA breaks. In order to explore the directionality of nucleosome sliding by ALC1, we adapted a fluorescence resonance energy transfer (FRET)-based nucleosome sliding assay[52] for monitoring nucleosome sliding at the single-molecule level[53]. We used a DNA construct comprising the 601 sequence as well as 12 bp of linker DNA and an end-positioned Cy5 FRET acceptor fluorophore on one side, and 78 bp on the other side (Fig. 3a) to assemble FRET-labeled nucleosomes. To monitor the ALC1-catalyzed remodeling of individual nucleosomes using single-molecule FRET (smFRET)[36,54–56], we immobilized them and recorded their fluorescence emissions (Fig. 3a). The nucleosomes exhibited an intermediate starting FRET value (~0.4) (Fig. 3b). A FRET decrease upon addition of ALC1 and ATP would indicate a nucleosome movement away from the short DNA end, while a FRET increase would indicate a movement towards the short end. To differentiate FRET decrease from photobleaching, we constantly monitored the presence of the acceptor fluorophore using alternating laser excitation[57]. Since the FRET signal exhibits non-monotonic behavior in relation to nucleosome position when a nucleosome approaches the short DNA linker, we limit our analysis to the direction of the initial movement.

Our previous results demonstrated that a constitutively active ALC1 mutant, where the mutation released the macrodomain from its autoinhibitory interaction with the motor, has a preference for sliding nucleosomes away from a short DNA linker[14]. To probe whether the same might be for wild-type ALC1 lacking this mutation, we used a PARylated histone peptide, H3(1-20)-(Ser10(ADPr)$_4$), to activate ALC1. Strikingly, when ADP ribose chains were provided in trans,

ALC1 demonstrated a preference for sliding nucleosomes towards rather than away from the short DNA linker (61% of traces with initial increase in FRET, Fig. 3c). Such sliding of nucleosomes towards and potentially past a DNA end would most likely be detrimental at sites of DNA damage. We therefore reasoned that the specific recruitment of ALC1 to one side of the nucleosome due to asymmetric PARylation could bias remodeling to slide the nucleosome away from the DNA linker closest to the ADP-ribosylation site. Thus we tested the remodeling directionality for nucleosomes harboring (ADPr)$_4$ chains attached to H2B Ser6. We combined oriented hexasomes with Cy3-H2A and non-ADP-ribosylated H2B or H2B-(Ser6(ADPr)$_4$) with wild-type or ADP-ribosylated dimer (H2A/H2B-(Ser6(ADPr)$_4$)) to obtain FRET-labeled nucleosomes with ADP ribose chains on either or both sides. Indeed, among nucleosomes featuring a single, site-specifically and homogeneously ADP-ribosylated dimer on the shorter linker-proximal side, a majority (57%) of individual traces featured an initial decrease in FRET. This is consistent with ALC1-induced sliding of nucleosomes away from the shorter linker, which notably differs from only 39% of such instances observed when PAR chains were provided in trans (Fig. 3c). Conversely, nucleosomes that featured the site-specific ADP-ribosylation on the longer linker-proximal side were preferentially shifted towards the shorter linker (46% of traces with initial FRET decrease versus 57% for the shorter-linker proximal side). When ADP ribose chains were present on both sides, ALC1 exhibited essentially no preference (51% of traces with initial FRET decrease) for movement in either direction (Fig. 3c). Interestingly, for both ALC1 and the closely related Chd1 remodeler, the remodeling initiation time was substantially shorter when the reaction was started by introducing the enzyme and ATP at the same time, compared to first allowing the enzyme to bind nucleosomes in the absence of ATP (Supplementary Fig. 4b). This suggests that in the absence of ATP, both ALC1 and Chd1 bind to nucleosomes in a long-lived off-pathway conformation.

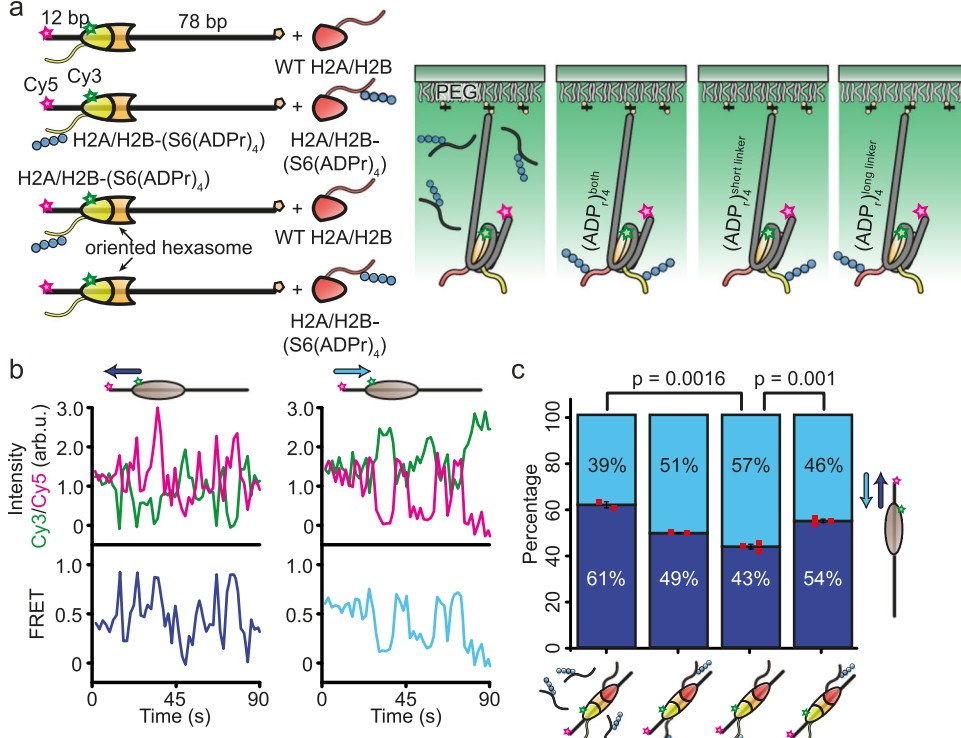

**Fig. 3 | The position of the ADP-ribosylation site relative to the nucleosome core biases the initial directionality of sliding. a** Schematics depicting the reconstitution of FRET-labeled, non-ADP-ribosylated nucleosomes or of FRET-labeled nucleosomes with H2B-Ser6(ADPr)$_4$ either on both sides, or on the longer-linker or shorter-linker DNA side only, and TIRF-based FRET detection. **b** Example Cy3 (green), Cy5 (magenta) fluorescence and FRET (blue) traces of non-ADP-ribosylated nucleosomes initially remodeled by ALC1 towards the shorter (dark blue) or longer linker DNA (light blue). 1 μM ADP-ribosylated H3 peptide H3(1-20)-(Ser10(ADPr)$_4$) was added to non-ADP-ribosylated nucleosomes to activate ALC1. Example traces for all conditions are presented in Supplementary Fig. 3a.

**c** Percentage of remodeling time traces that feature an initial increase (dark blue) or decrease (light blue) in FRET for non-ADP-ribosylated nucleosomes as well as for nucleosomes with H2B-Ser6(ADPr)$_4$ either on both sides, or on the longer-linker or shorter-linker DNA side only. 1 μM ADP-ribosylated H3 peptide H3(1-20)-(Ser10(ADPr)$_4$) was added to non-ADP-ribosylated nucleosomes to activate ALC1. Error bars indicate SEM ($n$ = 214, 423, 373 and 284 traces). Red squares represent the results from individual repeats. Two-sided t-test was used to compare the results (see Source Data file for additional details). No adjustment for multiple comparison was made. Source data are provided as a Source Data file.

Taken together, our data indicate that asymmetrically ADP-ribosylated nucleosomes were initially moved away from the linker closest to the ADP-ribosylation site, implying that its position relative to the nucleosome core biases the sliding directionality of ALC1.

## Discussion

Remodelers such as ALC1 engage the nucleosome at SHL2, an internal DNA site located ~20 bp from the dyad[37–40], to move DNA around the histone octamer. With respect to the SHL2 site of DNA translocation, one H2A/H2B dimer is on the entry side, where DNA is shifted onto the nucleosome, while the other dimer is located on the exit side[58]. The directionality of nucleosome sliding therefore depends on which side of the two-fold symmetric nucleosome the remodeler engages. Due to its symmetry, the nucleosome can in principle bind two remodelers at the same time. In such a scenario, each remodeler can reposition the nucleosome in a unidirectional manner only, where the action of remodelers on opposite sides of the nucleosome would give rise to sliding in opposing directions. The mechanisms underlying a potential competition or synergistic action of opposing remodelers are incompletely understood. Chd1 and other CHD-family remodelers feature a ChEx fragment that engages the exit-side acidic patch of the nucleosome, which has been proposed to block acidic patch-targeting remodelers poised to slide nucleosomes in the opposite direction[59]. In the case of ALC1, the mechanisms that govern the sliding directionality in the vicinity of DNA breaks remained unknown. Here, we examined the dependence of ALC1-induced remodeling on site-specific nucleosome PARylation. The directionality of ALC1-catalyzed nucleosome

sliding has not been entirely clear[8,9,14]. Our previous results with a constitutively active ALC1 mutant suggested a preference for centering nucleosomes. Surprisingly, here we observed an opposite preference when wild-type ALC1 was activated by PAR chains in trans. One potential explanation for these observations could be that the macro domain can interact with linker DNA via its PAR-binding site and act in a manner analogous to the DNA binding domain of Chd1[60,61]. However, in the presence of ADP-ribose chains engaging the ADP-ribose binding site, such an interaction with DNA is likely impossible, potentially resulting in an opposite remodeling directionality. Indeed, we have previously demonstrated that the macro domain of ALC1 can bind DNA[10]. The asymmetric introduction of PAR chains on the nucleosome could provide a means to simultaneously recruit and activate ALC1, and at the same time ensure preferential remodeling in the direction away from the closest DNA end. Indeed, our results demonstrate that PARP1/HPF1 preferentially PARylates histone tails closest to the DNA end. This in turn could direct ALC1 to engage the nucleosome such that it is preferentially shifted away from the DNA end. In fact, our single-molecule experiments with nucleosomes containing a single ADP-ribosylated H2B dimer on either side show that ALC1 preferentially slides nucleosomes away from the linker proximal to the ADP-ribosylated side (Fig. 3). Our data therefore revealed an unexpected bias in the sliding directionality as defined by the location of ADP-ribose chains relative to the nucleosome core. Interestingly, even a single minimal PAR chain enables ALC1 remodeling in both directions, albeit with different probabilities. Most likely, the enzyme can switch remodeling direction without completely dissociating from the

nucleosome. It is notable that ALC1 has a preference for sliding nucleosomes towards the short linker when PAR chains are added in trans, but has essentially no preference when sliding symmetrically PARylated nucleosomes. One of the main differences between these two cases is the fact that the macro domain movement is likely substantially constrained when PAR chains are attached to histones. These results suggest an intriguing possibility: that the directionality of ALC1 remodeling can be influenced by the reach of the macro domain.

In vivo, PARP1/HPF1-mediated ADP-ribosylation occurs on both histones H2B and H3[18]. Since we cannot generate nucleosomes with defined asymmetry in the H3/H4 tetramer[62], we chose H2B as a representative substrate to examine the effect of histone ADP-ribosylation on nucleosome sliding directionality. We note that asymmetry in H3 ADP-ribosylation is also likely to play a role in determining remodeling directionality in vivo.

Our cryo-EM structure of an ALC1-nucleosome complex obtained with a homogeneously ADP-ribosylated H2A/H2B dimer on one side shows that the ALC1 ATPase is recruited side-specifically to the SHL 2 location that is proximal to the ADP-ribosylated H2B tail (Fig. 2). We cannot formally rule out the possibility that the DNA linker length could contribute to such a binding preference. However, we consider such a scenario unlikely, since when ALC1 is activated by a PARylated peptide in trans, it has a preference for sliding nucleosomes towards rather than away from the shorter linker. This suggests that in the absence of nucleosome ADP-ribosylation, ALC1 has a preference for a binding orientation that is opposite to that observed in our cryo-EM structure. In agreement with our single-molecule analyses, such specific recruitment would bias the directionality of remodeling to slide the nucleosome away from the linker closest to the ADP-ribosylation site.

ADP-ribosylation by PARP1 is notoriously heterogeneous and can produce PAR chains of various lengths and branching, as well as on various acceptor proteins. In the context of chromatin at DNA lesions in vivo, the site- and degree-specific ADP-ribosylation is most likely tightly regulated to orchestrate spatiotemporal control over unique remodeler activities[30,63]. Given that HPF1 is required for the initiation of histone ADP-ribosylation but interferes with chain elongation[30,64,65], HPF1 must be present at sub-saturating concentrations to permit both processes. Consistent with this, the expression level of HPF1 was shown to be 20-fold lower than that of PARP1[35]. As a rate-limiting step, initiation is therefore a likely candidate for controlling catalytic output of the histone ADP-ribosylation reaction, which could be harnessed to establish directional nucleosome sliding in the vicinity of double-strand breaks. Indeed, we found that PARP1/HPF1 initially installs PAR chains predominantly on the double-strand break-proximal side of nucleosomes (Fig. 1). This result is consistent with a structure of PARP2 and HPF1 bound to a nucleosome[66], which suggests that the H3 and H2B tails closest to the active site are likely to be favored for modification.

Taken together, our data suggest a role for ALC1-induced directional nucleosome sliding during the processing of a DNA break: PARP1 or PARP2 would bind to the break and initially ADP-ribosylate the break-proximal side of the octamer. This in turn would recruit ALC1 side-specifically to the nucleosome and slide the ADP-ribosylated octamer away from the break, making it more accessible to repair factors (Fig. 4). Our results highlight an important role of HPF1-stimulated nucleosomal histone PARylation. Besides playing a critical role in promoting the efficient recruitment of ALC1 to and activation at lesion-proximal nucleosomes, histone PARylation now also appears to help ensure the required direction of sliding.

## Methods

### Expression and purification of recombinant proteins

The wild-type ALC1 construct (16-879) comprises the human 6xHis-tagged ALC1/CHD1L sequence, and it was expressed and purified as

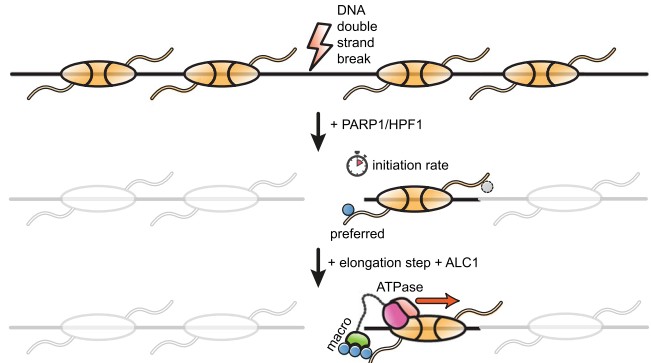

**Fig. 4 | Model for the mechanistic role of ALC1 upon recognition of an asymmetrically ADP-ribosylated nucleosome.** The PARP1/HPF1 complex is rapidly recruited to a DNA double-strand break, and initiates the formation of an oligo-ADP-ribose chain, preferentially on the proximal side of the nucleosome. Asymmetric PARylation, in turn, recruits ALC1 side-specifically to the nucleosome, with the ALC1 ATPase motor engaging the SHL2 proximal to the modified histone tail. The binding of the DNA break-proximal PAR chain to the ALC1 macro domain displaces it from the ATPase, releasing ALC1 remodeling activity that slides the histone core away from the DNA break. Such remodeling may facilitate chromatin relaxation in a DNA damage-specific chromatin context.

described in ref. 10. In short, protein overexpression in Rosetta 2 (DE3) cells (Novagen) was induced by adding IPTG at a final concentration of 0.5 mM to the culture media. After the cell harvest by centrifugation at $5000 \times g$ for 20 min, the cell pellet was lysed by sonication and subjected to additional centrifugation. As a first purification step, the supernatant was filtered and loaded onto a HisTrap HP 5 ml affinity column (Cytiva). Bound proteins were eluted with imidazole, and desired fractions were identified by SDS-PAGE and pooled. As a second purification step, the pooled ALC1 fractions were subjected to ion exchange chromatography on a tandem HiTrap Q HP 5 ml and HiTrap SP HP 5 ml (Cytiva), with a Q column to trap contaminating DNA and removed before eluting the protein from the SP column with a gradient to 1 M NaCl. In a third purification step, the pooled ALC1 fractions were concentrated and subjected to size-exclusion chromatography on a Superdex 200 16/60 column (GE Healthcare). The desired fractions after size exclusion chromatography were identified by SDS-PAGE, pooled, concentrated, flash frozen in liquid nitrogen, and stored in a −80 °C freezer for later use.

The plasmid for a human 6xHis-tagged PAPR1 (1-1014) was a kind gift from John M. Pascal, and it was expressed and purified as described in ref. 67. The PARP1 expression and purification were performed similarly to ALC1, with the following differences. For PARP1 expression, benzamide was added to the expression culture at a final concentration of 10 mM. For PARP1 purification, in a second purification step, the pooled fractions were subjected to a HiTrap Heparin HP 5 ml column (Cytiva).

The human 6xHis-tagged HPF1 was expressed and purified as described in ref. 68. The HPF1 expression and purification were performed similarly to ALC1, except the second purification step was omitted.

Chd1 (*S. cerevisiae*, residues 118−1274) was expressed in *E. coli* and purified as previously described[69,70]. Briefly, cell pellets were lysed on ice by sonication and addition of lysozyme, and after clarification by centrifugation, protein was purified by passage over a NiNTA resin followed by SP-FF. The 6xHis tag was removed by protease digestion overnight at 4 °C, and protein was further purified by repassage over the NiNTA resin and subjected to S200 size exclusion chromatography. The purified protein was concentrated and stored in small aliquots at −80 °C. Thawed aliquots of remodeler protein were always kept on ice and used within 12 h for each experiment.

## Labeling of histone H2A and H2B

Recombinant histones H2A K120C and H2B T120C (*Xenopus laevis*) were expressed and purified as previously described[56], except histone H2A K120C used in the $(ADPr)_4^{short\ linker}$ FRET-labeled nucleosome, which was purchased from the Histone Source Protein Expression and Purification Facility, Colorado State University, Fort Collins, CO. Briefly, BL21 (DE3) pLysS (Novagen) cells were used. Expression was induced at $OD_{600} = 0.6$-$0.8$ for 2 h at 37 °C with 1 mM IPTG. The pellets were resuspended in 40 mM NaOAc pH 5.2, 1 mM EDTA, 10 mM lysine, 200 mM NaCl, 5 mM β-mercaptoethanol, 6 M urea, and a protease inhibitor cocktail (Roche). Cells were lysed by sonication and centrifuged. Filtered supernatant (0.45 μm filter) was purified over a tandem 5 ml SP HP column (GE Healthcare) and 5 ml Q HP column (GE Healthcare) to trap contaminating DNA. A salt gradient was used for elution. Histone-containing fractions were collected and dialyzed overnight against cold water, supplemented with 15 mM Tris pH 8.0 and passed over a 5 ml Q HP column (GE Healthcare). Finally, histones were concentrated to 5 mg/ml, flash frozen, and stored at −80 °C.

For labeling, one milligram of purchased lyophilized histone protein was diluted in unfolding buffer (20 mM Tris pH 7.0, 7 M guanidine-HCl, 5 mM EDTA, 1.25 mM TCEP) and incubated for 2 h at room temperature in the dark. Cy3- or Cy5-maleimide was dissolved in DMSO and added to the protein at a final concentration of 0.75 mM. After 3 h in the dark at room temperature, the reaction was quenched with a final concentration of 80 mM β-mercaptoethanol. The labeled protein was dialyzed nine times against dialysis buffer (20 mM Tris pH 7.0, 7 M guanidine-HCl, 1 mM DTT) and then used in histone dimer assembly. The labeling efficiency of the Cy3- and Cy5-labeled histones was approximately 70–85%.

## Preparation of ADP-ribosylated histone H2B

The synthetic histone H2B peptide (1-16) with the sequence "PDPAK-SAPAPKKGSKK" and H3 peptide (1-20) with the sequence "ARTKQ-TARKSTGGKAPRKQL" and with C-terminal bis(2-sulfanylethyl)amido (SEA) groups were prepared using the peptide synthesis and MES-Na thioester preparation methods[71]. The peptides were characterized by LC/MS, modified by PARP1 and HPF1 in two enzymatic steps and purified to homogeneity using C18 semi-preparative RP-HPLC[30]. The H2B (1-16) peptides with tri- and tetra ADP-ribose on them were ligated to the N-terminal cysteine residue of a recombinant histone H2B fragment (H2B A17C, amino acids 17-125) using a native chemical reaction performed at 37 °C in a degassed buffer containing 6 M guanidine-HCl, 100 mM sodium phosphate dibasic, 20 mM TCEP and 100 mM TFET, at pH 7 for 3 h. Next, TFET removal and desulfurization were carried out, taking care to prevent exposure to air as it can cause oxidation of the histones[71]. The desulfurized full-length histone products were purified over a C18 semi-preparative RP-HPLC. The fractions were analyzed by analytical C18 RP-HPLC and ESI-MS. The pure fractions were pooled, lyophilized, and stored at −80 °C until use. A similar strategy was employed to prepare the ADP-ribosylated H3 histone. We used the H3(1-20)-(Ser10(ADPr)4) peptide as detailed previously[30].

All analytical RP-HPLC was performed on an Agilent 1260 series instrument with a Waters XBridge Peptide C18 column (5 μm, 4 × 150 mm) in 0.1% TFA in water (Solvent A) and 90% acetonitrile, 0.1% TFA in water (Solvent B) as mobile phases. All LC/MS analyses were performed in 0.1% formic acid in water (Solvent A) and 0.1% formic acid, 90% acetonitrile in water (Solvent B) as mobile phases. The mass spectrometer used for analysis is a single quadrupole Agilent LC/MSD. Analysis is of intact masses only, and traditional deconvolution of mass-to-charge peaks was used to calculate intact mass. A gradient of 0-90% Solvent B over 15 min was carried out for sample analysis on a 300 SB-C18 column (3.5 μm; 4.6 × 100 mm, Agilent Technologies). RP-HPLC and ESI-MS characterization of ADP-ribosylated histones is provided in Supplementary Fig. 5.

## Nucleosome assembly

The DNA fragments for nucleosome assembly were prepared following the previously described strategy[41]. Briefly, DNA was amplified from a 601 sequence-containing plasmid[32] by PCR in 96-well plates using Phusion polymerase, Phusion HF buffer (New England Biolabs) and each primer at 1 μM (Integrated DNA Technologies). Sequences of oligonucleotides used in this study are provided in the Supplementary Table 1. For FRET-labeled nucleosomes, the reverse primer (corresponding to the short linker end) was labeled with Cy5 while the forward primer (corresponding to the long linker end) was biotinylated. For labeled DNA, the amplified product was purified using a PrepCell (BioRad). For unlabeled DNA, the amplified product was then purified by anion exchange chromatography on a HiTrap Q HP 5 ml column (Cytiva) by loading 10 ml of pooled PCR reaction on the column at 1 ml/min in 50 mM Tris−HCl pH 8, 100 mM NaCl, 0.1 mM EDTA. Elution was performed with a gradient to 1 M NaCl over 20 column volumes. Adequate fractions were identified by native PAGE on a 10% poly-acrylamide gel, pooled, subjected to ethanol precipitation, and dissolved in a small volume of pure water.

Purified recombinant histones (from *Xenopus laevis*) were purchased from the Histone Source Protein Expression and Purification Facility, Colorado State University, Fort Collins, CO. The histone tetramer was refolded by mixing equimolar amounts of H3 and H4 dissolved in the unfolding buffer (20 mM Tris−HCl pH 7.5, 6 M guanidine HCl, 5 mM DTT) and dialyzing the mixture against refolding buffer (10 mM Tris−HCl pH 7.5, 2 M NaCl, 1 mM EDTA, 5 mM 2-mercaptoethanol) three times throughout 20 h. The resulting histone tetramer was concentrated and purified by size exclusion chromatography on a Superdex 200 16/60 column (GE Healthcare). Pure fractions were identified by SDS−PAGE[72,73]. The same refolding and purification protocol was followed for histone dimer, mixing histone H2A and H2B in equimolar amounts.

Oriented nucleosomes for cryo-EM and PARylation analyses were assembled using the strategy described in ref. 31. Firstly, hexasomes were formed by combining DNA, tetramer and limiting amounts of dimer in a 1:1.2:0.5 molar ratio in high-salt buffer (10 mM Tris−HCl pH 7.5, 2 M NaCl, 1 mM EDTA, 1 mM DTT) and dialyzing continuously to 0 M NaCl[72,73]. Hexasomes were purified over a 7% native acrylamide column (60:1 acrylamide:bisacrylamide) using a MiniPrep Cell (BioRad) apparatus. Complete nucleosomes were assembled before the experiment by adding H2A/H2B dimer to hexasomes in 2-fold molar excess and incubating at 37 °C for 15 min. Oriented nucleosomes for smFRET experiments were assembled as described in ref. 34. Briefly, hexasomes were first reconstituted by salt gradient dialysis on a truncated version of the Widom 601 positioning sequence ("core") that is too short to enable nucleosome formation. Next, a biotinylated "stem" DNA piece was ligated to the hexasomes (1.125x excess of the "stem", T4 DNA ligase, 30 min at 16°C in 10 mM Tris, pH 7.5, 0.6 mM MgCl, 0.1 mM ATP, 1 mM DTT). Finally, complete nucleosomes were assembled before the experiment by adding H2A/H2B dimer to hexasomes in 2-fold molar excess and incubating at 37°C for 15 minutes.

## Single-molecule FRET assay

The DNA construct for smFRET assay comprises the Widom 601 nucleosome positioning sequence, 12 bp of linker on one side, which is 5′-labeled with Cy5 and 78 bp linker on the other side, which is 3′-biotinylated. The FRET nucleosomes were assembled in the following steps. Firstly, the oriented hexasomes were assembled from Cy5-labeled and biotinylated DNA, histone tetramer and H2A(K120C-Cy3)/H2B dimer[34]. This allowed for the controlled incorporation of the Cy3 label in a specific orientation (Fig. 3a). The hexasomes were purified over 7% polyacrylamide gel using the MiniPrepCell (BioRad) apparatus.

The initial remodeling directionality was measured as described in ref. 14. In brief, the biotinylated FRET-labeled nucleosomes were

immobilized on a PEG (poly[ethylene glycol])-coated quartz slide saturated with streptavidin[36]. Cy3 and Cy5 fluorophores were excited with 532 nm Nd:YAG and 638 nm diode lasers, respectively, and fluorescence emissions from Cy3 and Cy5 fluorophores were detected using a custom-built prism-based TIRF microscope. To check the presence of an intact donor fluorophore, the sample was alternately excited with 532 nm and 638 nm lasers during the experiment. Data acquisition was controlled by MicroManager[74]. Data were analyzed using custom scripts in the Fiji distribution of ImageJ[75], IDL, and MATLAB[56,76]. Remodeling experiments were carried out in the imaging buffer containing 40 mM Tris pH 7.5, 12 mM HEPES pH 7.9, 60 mM KCl, 0.32 mM EDTA, 3 mM MgCl$_2$, 100 mg/mL acetylated BSA (Promega), 10% (v/v) glycerol, 10% (w/v) glucose, supplemented with 2 mM Trolox to reduce photoblinking of the dyes (Rasnik et al., 2006), as well as an enzymatic oxygen scavenging system (composed of 800 µg/ml glucose oxidase and 50 µg/ml catalase). Using a syringe pump (Harvard Apparatus), remodeling was initiated by injecting the imaging buffer supplemented with 1 µM ALC1 and 1 mM ATP/MgCl$_2$. The initial remodeling direction and remodeling initiation times were determined by visually inspecting the FRET traces. Bar graphs were plotted using Origin.

## SDS-PAGE gel assay

The DNA construct for the SDS-PAGE assay comprises the Widom 601 nucleosome positioning sequence, 3 bp of linker on one side and 78 bp linker on the other side. The nucleosomes were assembled as follows. DNA, histone tetramer and H2A/H2B (T120C-Cy3) dimer were mixed to form hexasomes with Cy3-labeled dimer in a specific orientation, facing the 3 bp linker DNA. After the purification, hexasomes were mixed with H2A/H2B (T120C-Cy5) dimer to form oriented nucleosomes (Fig. 1a).

Oriented Cy3- and Cy5-labeled nucleosomes (200 nM final concentration) were first mixed with PARP1 and HPF1 in 8-fold and 20-fold access, respectively, in a reaction buffer (25 mM HEPES-NaOH pH 8, 50 mM NaCl, 0.1 mM EDTA and 1 mM DTT). After one hour of incubation on ice, NAD$^+$ was added to each reaction at concentrations ranging from 0.5 to 32 µM in a final volume of 6 µl. The PARylation was allowed for 30 minutes on ice. After the incubation time, SDS loading buffer (without bromophenol blue) was added. The proteins were denatured for 5 min at 95 °C. Reactions were loaded onto an 18% polyacrylamide gel (with 6% stacking gel) and run for 48 hours at 2-4 W in 1x SDS running buffer. Gels were imaged on a BioRad ChemiDoc MP Imaging System using ImageLab software. The Cy3 and Cy5 intensities were measured using Fiji (ImageJ)[74] and converted to PARylation extent in Excel.

## Cryo-EM sample preparation

The DNA construct for cryo-EM comprises the Widom 601 nucleosome positioning sequence, 0 bp of linker on one side and 10 bp linker on the other side. The uniquely ADP-ribosylated nucleosomes were assembled as follows. DNA, histone tetramer and wild-type H2A/H2B dimer were mixed to form a hexasome. After hexasome purification, the H2A/H2B(S6(ADPr$_3$)) was added to form the modified nucleosome.

The mixture of modified nucleosomes at 1 µM, ALC1$^{fl}$ at 6 µM, and ADP-BeF$_3$ at 1 mM (1 mM ADP, 3 mM BeSO$_4$, 15 mM NaF and 1 mM MgCl$_2$) was incubated for 60 minutes on ice before vitrification. ADP-BeF$_3$ was prepared as a 10x stock freshly before use. Quantifoil R 1.2/1.3 Cu 200 grids (Electron Microscopy Sciences) were glow-discharged at 30 mA and 0.4 mbar with negative polarity for 120 s using a PELCO easiGlow glow discharger. A volume of 3 µl of the sample was applied onto grids and immediately blotted for 4 s. Grids were plunge-frozen into liquid ethane using a Vitrobot Mark IV (Thermo Fisher Scientific) operated at 100% relative humidity and 4 °C.

## Cryo-EM data collection and image processing

Cryo-EM data were collected at the SciLifeLab facility in Stockholm, Sweden, on a Titan Krios equipped with a Gatan K3 BioQuantum detector operated in counting mode with an energy filter slit width of 20 eV. Magnification was 105 kx, resulting in an image pixel size of 0.862 Å/pixel. A total accumulated dose of 37.6 e−/Å$^2$ was fractionated in 40 movie frames. A total of 22 155 movies were collected from two replicate grids prepared on the same day from the same batch of sample.

Raw movies were motion-corrected using UCSF MotionCor2 version 1.3.2[77], and CTF parameters were estimated using CTFFIND4 version 4.1.9[78], both from within RELION version 3.1.3. Particle-picking was done using Topaz version 0.2.4[79] and identified 4 128 225 particles.

A subset of 739 042 particles was retained after several rounds of reference-free 2D classification and 3D classification in RELION (the initial reference for 3D classification was an ab initio 3D model

**Table 1 | Cryo-EM data collection, refinement and validation statistics**

|  | ALC1 / Nuc-H2B-ADPr3 (EMDB-15777) (PDB 8BOA) |
|---|---|
| **Data collection and processing** |  |
| Magnification | 105,000× |
| Voltage (kV) | 300 |
| Electron exposure (e−/Å$^2$) | 37.6 |
| Defocus range (µm) | −1.0 to −2.5 |
| Pixel size (Å) | 0.862 |
| Symmetry imposed | C1 |
| Initial particle images (no.) | 4,128,225 |
| Final particle images (no.) | 212,256 |
| Map resolution (Å) | 3.0 |
| FSC threshold | 0.143 |
| Map resolution range (min/mean/max, Å) | 1.9/2.8/5.0 |
| **Refinement** |  |
| Initial model used (PDB code) | 7OTQ |
| Model resolution (Å) | 3.0 |
| FSC threshold | 0.143 |
| Model resolution range (min/mean/max, Å) | 1.9/2.8/5.0 |
| Map sharpening B factor (Å$^2$) | −117.4 |
| Model composition |  |
| Non-hydrogen atoms | 15,843 |
| Protein residues | 1222 |
| DNA residues | 298 |
| B factors (Å$^2$) |  |
| Protein | 79 to 343 |
| DNA | 0 to 768 |
| R.m.s. deviations |  |
| Bond lengths (Å) | 0.008 |
| Bond angles (°) | 0.956 |
| Validation |  |
| MolProbity score | 0.98 |
| Clashscore | 0.99 |
| Poor rotamers (%) | 0.58 |
| Ramachandran plot |  |
| Favored (%) | 96.93 |
| Allowed (%) | 2.66 |
| Disallowed (%) | 0.42 |

generated from the data). These particles were analyzed with the 3D variability analysis (3DVA) protocol in cryoSPARC version 3.3.1[80], solving for three principal components (all other parameters were left to their defaults). The 3DVA results were clustered into two groups, one of which contained 212,256 particles and showed sharp secondary structure features in the region of ALC1. These particles were subjected to non-uniform refinement in cryoSPARC[81], which reached a global resolution of 3.0 Å. We estimated the occupancy of the features in the resulting map using OccuPy version 0.1.13[51]. The map from non-uniform refinement was also post-processed with deepEMhancer[82], using the "highRes" weights, to facilitate model building and visualization.

The atomic model from PDB entry 7OTQ (Cryo-EM structure of ALC1/CHD1L bound to a PARylated nucleosome)[13] was manually refined in ISOLDE version 1.4[83] against the map from non-uniform refinement, with the deepEMhancer map used as a visual aid only (not used to guide the molecular dynamics flexible fitting). The model was finally subjected to real-space refinement against the non-uniform refinement map, using phenix.real_space_refine version 1.20.1-4487[84] with the parameter file generated by ISOLDE. Data collection and model refinement statistics are shown in Table 1.

All figures were prepared with ChimeraX version 1.4[85], using the deepEMhancer map and the refined atomic model, except for the local resolution figure (Supplementary Fig. 2a) and the occupancy figure (Supplementary Fig. 3), which used the map from non-uniform refinement.

### Reporting summary
Further information on research design is available in the Nature Portfolio Reporting Summary linked to this article.

## Data availability
Raw movies, particle coordinates and extracted particles of the final set of particles were deposited in the EMPIAR database with accession code EMPIAR-11211 (Single-particle cryo-EM dataset of ALC1 bound to an asymmetric, site-specifically PARylated nucleosome). Maps and masks were deposited in the EMDB with accession code EMD-15777 (Cryo-EM structure of ALC1 bound to an asymmetric, site-specifically PARylated nucleosome). The atomic model was deposited in the PDB with accession code 8B0A (Cryo-EM structure of ALC1 bound to an asymmetric, site-specifically PARylated nucleosome). Cryo-EM structure of ALC1 bound to an enzymatically PARylated nucleosome is available in the PDB with accesion code 7OTQ (Cryo-EM structure of ALC1/CHD1L bound to a PARylated nucleosome). Single-molecule FRET data are available in the SciLifeLab Data Repository [https://doi.org/10.17044/scilifelab.24764697] (Asymmetric nucleosome PARylation at DNA breaks mediates directional nucleosome sliding by ALC1)[86]. Source data are provided with this paper.

## Code availability
Custom MATLAB codes used to analyze the single-molecule FRET data are available in the SciLifeLab Data Repository [https://doi.org/10.17044/scilifelab.24764697] (Asymmetric nucleosome PARylation at DNA breaks mediates directional nucleosome sliding by ALC1)[86].

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

## Acknowledgements

Cryo-EM data were collected at the Swedish National Cryo-EM Facility, SciLifeLab, Stockholm University, and Umeå University. We thank M. Carroni, D. Morado, K. Wallden, and J. Conrad for assistance with cryo-EM data collection. We thank J. Pascal for generously providing the PARP1 expression vector. The HPF1 expression vector was a kind gift from K. Luger. G.L. is funded by the National Institute of General Medical Science (1R35GM147140), the Welch Foundation (I-2039-2020040), the American Heart Association (937595), and the Cancer Prevention and Research Institute of Texas (RR180051). S.D. is supported by the European Research Council (ERC Starting Grant 714068), the Knut and Alice Wallenberg Foundation (Grant KAW 019.0306), the Swedish Research Council (VR Grant 2019–03534), and Cancerfonden (Grant 19 0055 Pj). S.D. is an EMBO Young Investigator.

## Author contributions

J.M. prepared ADP-ribosylated histones and H3 tail peptide. L.B., G.M., and A.S. collected the ALC1 smFRET data, and L.B., and G.M. analyzed them with help from A.S. L.B. prepared samples for cryo-EM and vitrified grids. L.B. and G.G. collected, processed and analyzed cryo-EM data. G.G. built and refined the atomic model. L.B. and K.B. prepared smFRET nucleosome constructs. L.B. conducted the gel shift experiments. M.P. collected and analyzed the Chd1 smFRET data. L.B., G.G., A.S., and S.D. wrote the manuscript, with input from all the other authors. S.D. and G.L. acquired funding. S.D. oversaw the study.

## Funding

## Competing interests

The authors declare no competing interests.
