## [Peer Review File · Nature Communications]

Asymmetric nucleosome PARylation at DNA breaks mediates directional nucleosome sliding by ALC1Reviewer #1 (Remarks to the Author):

This manuscript studies the important DNA-damage response system of PARP1. After damage, addition of polyADP-ribose (PAR) by PARP1 is an important signal, and the authors investigate how this signal is deposited on histones near an artificial break. They use a clever assay where two copies of H2B are differently fluorescently labeled on two sides, and see that PAR modifications are different (Fig 1). One side modifies faster, but other side shows 2 (?) modifications. It is unclear what the difference is. Also, both sides are modified, so it is not a black/white answer.

The authors solve a cryo-EM structure of the ALC1 remodeler bound to a nucleosome with PAR asymmetric on histones. Different than before, ALC1 is on one side only, which the authors say is the side with the engineered PAR. This gives evidence that PAR influences where the remodeler binds.

In the final experiment, the authors make single-molecule measurements with engineered PAR nucleosomes. They look at sliding to the 12 bp end or the longer 78 bp side. From the example given, the movement to 12 bp end (FRET) is followed by low Cy5 – is that photobleaching? If so, is that also happening with movement to 78 bp end? This assay does not seem to clearly show nucleosome movement to the 78 bp side. The graph in Fig 3C is somewhat confusing also. The nucleosomes with PAR on different sides do not move significantly differently. Most different is the nucleosome with PAR on both sides. Could this be because the nucleosome with both side PARylated was made differently? Do the authors know that the PAR modification they engineer is the same as what PARP1 makes?

Although the biochemistry in Fig1 is intriguing, the result in Fig 3 is not clear. Therefore, one of the main conclusion of the authors does not seem right given the data that they show. The work has potential for an exciting story but current conclusions do not match all the data.

Reviewer #2 (Remarks to the Author):

In this paper, Bacic et al. used biochemical, cryo-EM, and single-molecule FRET assays to study the effects of nucleosome PARylation on the directionality of ALC1-mediated nucleosome sliding. Using an elegant strategy to generate asymmetrically modified nucleosomes, the authors showed that ADP-ribosylation by PARP1/HPF1 is more efficient on histones closer to the DNA end, that ALC1 preferentially binds to the ADP-ribosylated side of the nucleosome, and that the location of ADP-ribosylation relative to the nucleosome core biases the remodeling directionality of ALC1. Based on these findings, they propose a model in which ADP-ribosylation of histones close to a DNA double-strand break recruits ALC1 to the break-proximal side of the nucleosome and induces sliding of the octamer away from the break, thereby facilitating repair.

This is a very interesting study that provides important insights into the role of nucleosome PARylation in chromatin remodeling and DNA damage response. The data are of high originality and quality, and the manuscript is clearly written. Therefore, publication is recommended. I have a few comments that could be addressed by clarifications and discussions.

1. Different ADP-ribose chains were used in this study (e.g., ADPr3 in cryo-EM, ADPr4 in single-molecule FRET). Can the authors explain how these choices were made? It seems that the nature of the PAR chain could influence the ALC1 activity (see below.)

2. What is the general PARylation state (albeit heterogeneous) of the nucleosome using the enzymatic method? Figure 1 suggests that they are either mono or di-ADP-ribosylated, but the cartoon in Figure 3 indicates longer and/or branched products.

3. The example FRET trace in Figure 3b where the signal increased initially was followed by a decrease. Is this observed in other traces as well? If so, what is the explanation for it? The authors should show more example traces and perhaps perform more detailed analysis of the rich FRET data to provide more insights.

4. In Figure 3c, the distribution of FRET increase/decrease for the first column is expected to be somewhere between those for the second and third columns, because the ADP ribosylation on both sides of the nucleosome can induce sliding in opposite directions. However, the authors observed the highest fraction of FRET decrease of all conditions. This could be due to the different lengths of the PAR chain as mentioned above, or PARylation of H3 as the authors alluded to. Please discuss this point.

5. In the cryo-EM experiment, PARylation was installed on the longer-linker side, which seems counterintuitive because the paper argues that the shorter linker is the preferred side for PARP1/HPF1. Can the authors explain why this choice was made?

6. Line 127: "ALC1 is bound to the strong side (Figure 2c)..." Is this a typo (should be the "weak" side)?

7. The "short" and "long" linkers also have different meanings in different assays (Figure 1: 3/78; Figure 2: 0/10; Figure 3: 12/78). Is there a rule of thumb as to how short the linker needs to be in order to constitute a preferred site for ADP ribosylation?

Reviewer #3 (Remarks to the Author):

ALC1 is a single subunit chromatin remodeling enzyme that plays roles in DNA double strand break repair. ALC1 contains a Snf2-like ATPase domain, as well as a novel Macro domain that is known to bind to PAR chains that are attached to histones H3 or H2A, catalyzed by PARP enzymes. Previous studies from the authors' group and others, have shown that PARylation of histones enhances ALC1 activity by releasing the macro domain from an association with the ATPase lobes, and ALC1 prefers to slide an end-positioned nucleosome to the center (away from the ends). Furthermore, Deindl and colleagues published a 4 angstrom structure of ALC1 bound to a PARylated nucleosome (Bacic et al., 2021 ELIFE). This work also used a novel strategy to create nucleosomes where PARP enzymes preferred to modify histone sites adjacent to a short DNA end, mimicking the predicted modification of nucleosomes adjacent to a DSB (DNA contained a 5' phosphate on the short end, and a 5'-OH on the long end to control PARP activity). Ensemble FRET sliding assays with this substrate showed that PARylation could stimulate the ability of ALC1 to slide nucleosomes away from the DNA end.

In this current manuscript, Deindl and colleagues have extended their previous study by creating more homogeneous nucleosome substrates, combining an oriented hexosome assembly strategy and native ligation. In this way, they created nucleosomes where a single ADP modification was positioned on the histone H2A located adjacent to a short stretch of linker. Like their previous work, this substrate is predicted to mimic an *in vivo* location for PAR chains next to a DSB. Conversely they also made a substrate with the ADP on the opposite face of the nucleosome. Using an smFRET approach, the authors present evidence that the location of the PAR chain can influence the direction of nucleosome sliding by ALC1. Using one of these substrates, the authors also present a better cryoEM structure of ALC1 bound to a PARylated nucleosome (3 angstrom). But like previous structures, neither the PAR or the Macro domain of ALC1 are visualized.

In general, this work represents an incremental advance of our understanding of ALC1 action. The work confirms the major conclusions from their 2021 paper, perhaps with better reagents. This work reinforces the view that PARylation can direct ALC1 to bind to a preferred SHL2 position, and as expected this dictates the direction of sliding.

Specific comments:

Figure 2. The figure should indicate where the long and short linkers are located with respect to the bound ALC1 and where the ADP is predicted to be located.

Figure 3C. Here the authors need to show the data for the unmodified nucleosome substrate. They cite their older work, but that used a constitutively activated version of ALC1. The side-by-side comparison is essential here. The substrate is shown, the data should be available.

Figure 3. In previous work, the authors showed by ensemble FRET studies that an asymmetrically modified nucleosome yielded biphasic remodeling kinetics, as a function of ALC1 concentrations. At low concentration, faster rates were found, then higher concentrations revealed a slower rate of sliding. This was interpreted as ALC1 having different binding affinities for the two SHL2 locations – one location had higher affinity due to the presence of the PAR group on H3/H2A. Similar studies should be shown here. The expectation is that ALC1 will also occupy the SHL2 position that is not directed by ADP. The difference between the SHL2 occupancies is key, as the cryoEM conditions were established with an excess of ALC1 per nucleosome.

Intro, line 65 and 68. These two sentences seem incompatible. Are the authors referring to the 4 angstrom structure as not “high-resolution”? Perhaps this could be expanded. The current “high-resolution” structure has allowed the visualization of the DNA bases, but not much else.

In their previous work, the authors stated “The structure of PARP2 and HPF1 bound to nucleosomes indicates that, of the two H3 tails in a nucleosome, the one on the side of the DNA end bound by PARP2 is closest to its active site (Bilokapic et al., 2020). Target residues for ADP-ribosylation (Ser in KS motifs) in this proximal H3 tail should therefore be favored over the equivalent residues in the distal H3 tail.” Doesn’t this predict the data shown in Figure 1? Perhaps this should be mentioned here as well?

Based on the cryoEM data, the authors conclude that the location of the ADP-H2A adduct dictates which SHL2 is bound by ALC1. However, in the absence of a nucleosomal substrate with the label on the opposite face, it remains a possibility that the short linker is the determinate, not ADP.

In the smFRET studies, traces where the nucleosome is moved towards the edge (increased FRET) shows only a transient increase, followed by decreased FRET. This would imply that ALC1 rapidly re-orientes to the other SHL2? Is it still bound to PAR? Are the rates for this subsequent decrease slower than the initial movement? If the macro domain releases PAR when bound to the other SHL2, one would assume inhibition?

We thank the reviewers for their insightful comments and suggestions. We believe that they have helped us to considerably improve the manuscript. In particular, the reviewers' comments have prompted us to carry out a number of additional experiments and new analyses.

Moreover, we have repeated all of our smFRET experiments to obtain better statistics. In the process of doing so, we have noticed that the number of observed remodeling events was much larger when the reaction was initiated by adding ALC1 and ATP at the same time, as opposed to first allowing ALC1 to bind nucleosomes in the absence of ATP. We have now quantified the initiation time for remodeling under these two conditions. Notably, the initiation time was approximately 3-fold longer when ALC1 was preincubated with nucleosomes. Moreover, we observed the same trend with the closely-related Chd1 remodeler. This observation is unexpected, given that when the enzyme is added with ATP, the initiation might be limited by enzyme binding. One would expect shorter initiation times upon pre-incubation of the remodeler with nucleosomes. However, we observed the opposite effect, which may indicate that in the absence of ATP, both ALC1 and Chd1 bind to nucleosomes in an off-pathway conformation that persists for a relatively long time (tens of seconds) even after ATP is provided. We believe that this result is both mechanistically intriguing and of practical use for future research. We have therefore included the characterization of the remodeling initiation time in Figure S4b of the revised version of the manuscript, and added the following text to its results section:

“Interestingly, for both ALC1 and the closely related Chd1 remodeler, the remodeling initiation time was substantially shorter when the reaction was started by introducing the enzyme and ATP at the same time, compared to first allowing the enzyme to bind nucleosomes in the absence of ATP (Figure S4b). This suggests that in the absence of ATP, both ALC1 and Chd1 bind to nucleosomes in a long-lived off-pathway conformation”.

As part of our effort to generate more statistics for all smFRET experiments, we have also repeated the experiment with enzymatically-PARylated nucleosomes. While all smFRET experiments demonstrated excellent reproducibility throughout, in stark contrast, the remodeling of enzymatically-PARylated nucleosomes displayed large variability between independent experiments with different nucleosome preparations. This became evident only as part of an overall effort to increase statistics for our smFRET analyses. We do not know the underlying reasons, and we can only surmise that the heterogeneity and complexity of the PARylation reaction might amplify subtle variations in conditions (such as, temperature or incubation time) to substantially impact the length, branching, and distribution of PAR chains. Identifying the sources of variation in this experiment would require considerable additional time and effort. We therefore wish not to include the experiment with enzymatically-PARylated nucleosomes in the revised version of the manuscript and apologize to the reviewers for any confusion this may have caused.

The much-higher consistency of experiments with homogeneously-PARylated histones highlights the utility of this approach. Importantly, following the reviewers' suggestions, we have now prepared nucleosomes with both copies of H2B homogeneously PARylated and analyzed their remodeling by ALC1. We have further generated H3(1-20)-(Ser10(ADPr)₄) peptide for ALC1 activation in trans. The corresponding results are now included in the manuscript and discussed in detail below.

We have also used a recently published algorithm OccuPy (PMID 37726277) to quantify the occupancy of the features of our cryo-EM map. This new analysis revealed that the acidic

patch on the entry side contains additional density, while the acidic patch on the other side does not, which reinforces our previous observation that the entry-side acidic patch is required for ALC1-catalyzed nucleosome sliding (PMID 33357431). We have thus added a new supplementary Figure S3 and updated the results section to incorporate this new analysis:

“In addition, quantifying the occupancy of this map’s features with the recently published algorithm OccuPy (PMID 37726277) revealed that the acidic patch on the entry side contains additional density, while the acidic patch on the other side does not (Figure S3). This is consistent with our previous biochemical data, which showed that an interaction between the linker of ALC1 and the entry-side acidic patch is required for ALC1-catalyzed nucleosome sliding (PMID 33357431).”

Detailed point-by-point response to the reviewers’ comments

Reviewer #1:

This manuscript studies the important DNA-damage response system of PARP1. After damage, addition of polyADPribose (PAR) by PARP1 is an important signal, and the authors investigate how this signal is deposited on histones near an artificial break. They use a clever assay where two copies of H2B are differently fluorescently labeled on two sides, and see that PAR modifications are different (Fig 1). One side modifies faster, but other side shows 2 (?) modifications. It is unclear what the difference is. Also, both sides are modified, so it is not a black/white answer.

Next to the unmodified H2B band (bottom band), the gel in Figure 1b features additional bands corresponding to the mono-, di-, and tri-ADP-ribosylated H2B species, with the ‘higher-order’ bands mostly discernible for long linker-proximal H2B-Cy5. In other words, for the short linker-proximal dimer, the initiation is faster and therefore the unmodified H2B species is diminished at lower NAD⁺ concentrations. At the same time, elongation is faster for the other, long linker-proximal dimer, resulting in more pronounced higher-order bands for this dimer.

Since the HPF1 concentration in the cell has been shown to be 20-fold lower compared to PARP1 (PMID 27067600), initiation is most likely the rate-limiting step in PARylation. Our results suggest that in the cell, the probability of encountering PAR chains on the DNA break-proximal side of a nucleosome is several times higher than that for the distal side, which is expected to result in a directional remodeling bias away from the DNA break.

We have now added the following sentence to the results section to clarify this point:

“The concentration of HPF1 in the cell is 20-fold lower than that of PARP1, and HPF1 is required for initiation, but not elongation of histone PARylation (PMID 27067600). Initiation therefore likely represents the rate-limiting step for histone PARylation in vivo. Thus, the observed preference for initiating PARylation on the double-strand break-proximal H2B tail is expected to result in the installation of ADP-ribose chains mostly on one side of the nucleosome”.

The authors solve a cryo-EM structure of the ALC1 remodeler bound to a nucleosome with PAR asymmetric on histones. Different than before, ALC1 is on one side only, which the authors say is the side with the engineered PAR. This gives evidence that PAR influences where the remodeler binds.

In the final experiment, the authors make single-molecule measurements with engineered PAR nucleosomes. They look at sliding to the 12 bp end or the longer 78 bp side. From the example given, the movement to 12 bp end (FRET) is followed by low Cy5 – is that photobleaching? If so, is that also happening with movement to 78 bp end? This assay does not seem to clearly show nucleosome movement to the 78 bp side.

We thank the reviewer for emphasizing the importance of distinguishing photobleaching from a rapid decrease in FRET. During our smFRET experiments, we constantly monitor the presence of the acceptor using alternating laser excitation (i.e., by directly exciting the acceptor rather than the donor every other frame), so we can unambiguously distinguish a decrease in FRET from acceptor photobleaching. To clarify this, we have now added the following sentence to the results section:

“To differentiate FRET decrease from photobleaching, we constantly monitored the presence of the acceptor fluorophore using alternating laser excitation (PMID 15175430)”.

With DNA progressively moving into the nucleosome from the short linker side, the FRET efficiency is expected to initially increase but eventually reach maximum before decreasing again. For this reason, when observing an increase in FRET followed by a decrease (as in the example trace), we cannot be certain whether it arises from a reversal in remodeling direction after reaching high FRET values, or from processive movement towards the short linker. For remodeling away from the short-linker side, the signal is monotonous, however, such that a FRET decrease followed by a FRET increase represents a reversal in remodeling direction either within the same binding event, or with a new ALC1 molecule binding from solution. Indeed, we often observe such behavior, as evident from the new example traces we have now added (Figures 3b and S4a of the revised version of the manuscript). We now clarify this aspect with the following sentence in the results section:

“Since the FRET signal exhibits non-monotonic behavior in relation to nucleosome position when a nucleosome approaches the short DNA linker, we limit our analysis to the direction of the initial movement”.

The graph in Fig 3C is somewhat confusing also. The nucleosomes with PAR on different sides do not move significantly differently. Most different is the nucleosome with PAR on both sides. Could this be because the nucleosome with both side PARylated was made differently? Do the authors know that the PAR modification they engineer is the same as what PARP1 makes? Although the biochemistry in Fig1 is intriguing, the result in Fig 3 is not clear. Therefore, one of the main conclusion of the authors does not seem right given the data that they show. The work has potential for an exciting story but current conclusions do not match all the data.

We thank the reviewer for pointing this out. Based on the reviewers' comments, we have now carried out several additional new experiments. Most strikingly, we found that wild-type ALC1 activated by the addition of PAR chains in trans (using the H3(1-20)-(Ser10(ADPr)₄) peptide) has a preference for sliding nucleosomes towards rather than away from the short DNA linker. We envision such activity likely to be detrimental at sites of DNA damage. We believe that this experiment establishes the baseline for ALC1 remodeling behavior in the absence of nucleosome ADP-ribosylation. The incorporation of a PAR chain near the short DNA linker side therefore reverses the remodeling direction, likely in line with the requirements for DNA repair.

We completely agree with the reviewer that the preference for sliding away from the short DNA linker is not absolute. However, we believe the observed directional bias to be sufficient for producing an overall movement away from the short linker DNA. Importantly, this is clearly not the case in the absence of nucleosome PARylation, where we observed a clear bias for remodeling in the opposite direction, towards short linker DNA. We note that the difference in directionality between PAR chains on the short-linker side and PAR chains added in trans is statistically highly significant (two-sided $p=0.0016$), as is that for PAR chains on different sides of the nucleosome (two-sided $p=0.0009$).

As outlined above, we consider it more prudent to omit the results previously obtained with enzymatically-PARylated nucleosomes. While all smFRET experiments demonstrated excellent reproducibility throughout, in stark contrast, the remodeling of enzymatically-PARylated nucleosomes displayed large variability between independent experiments with different nucleosome preparations. We do not know the underlying reasons, and we can only surmise that the heterogeneity and complexity of the PARylation reaction might amplify subtle variations in conditions (such as, temperature or incubation time) to substantially impact the length, branching, and distribution of PAR chains. Identifying the sources of variation in this experiment would require considerable additional time and effort. We believe that the newly added experiments with exclusively homogeneously-PARylated histones allow for a much clearer analysis of remodeling directionality.

We have now updated the text and Figures 3 and S4 of the revised version of the manuscript to incorporate these new results and the measures of statistical significance.

Reviewer #2 (Remarks to the Author):

In this paper, Bacic et al. used biochemical, cryo-EM, and single-molecule FRET assays to study the effects of nucleosome PARylation on the directionality of ALC1-mediated nucleosome sliding. Using an elegant strategy to generate asymmetrically modified nucleosomes, the authors showed that ADP ribosylation by PARP1/HPF1 is more efficient on histones closer to the DNA end, that ALC1 preferentially binds to the ADP-ribosylated side of the nucleosome, and that the location of ADP ribosylation relative to the nucleosome core biases the remodeling directionality of ALC1. Based on these findings, they propose a model in which ADP ribosylation of histones close to a DNA double-strand break recruits ALC1 to the break-proximal side of the nucleosome and induces sliding of the octamer away from the break, thereby facilitating repair.

This is a very interesting study that provides important insights into the role of nucleosome PARylation in chromatin remodeling and DNA damage response. The data are of high originality and quality, and the manuscript is clearly written. Therefore, publication is recommended. I have a few comments that could be addressed by clarifications and discussions.

1. Different ADP-ribose chains were used in this study (e.g., ADPr₃ in cryo-EM, ADPr₄ in single-molecule FRET). Can the authors explain how these choices were made? It seems that the nature of the PAR chain could influence the ALC1 activity (see below.)

We thank the reviewer for this comment and apologize for the oversight of not having explained our choice of ADP-ribose chain lengths. It was previously shown that ALC1 requires an ADP-ribose chain length of at least three ADP-ribose units for efficient activation (PMID 29220653). Moreover, chains of three and four ADP-ribose units were shown to have virtually identical affinities for binding to the ALC1 macro domain, and essentially the same activating effect on ALC1 remodeling (PMID 34874266). The chemoenzymatic protein synthesis strategy that we used to generate histones with defined PAR modifications produces similar amounts of peptides with three and four ADP-ribose units that we separated by HPLC. Given that site-specifically PARylated histones represent the bottleneck for making nucleosomes and the virtually identical behavior of trimers and tetramers, we decided to make use of both for economic reasons. We used (ADPr)₃ with reduced chain flexibility for structural studies and (ADPr)₄ for smFRET analysis, where the conformational heterogeneity of ADP-ribose chains is less of a concern. We note that (ADPr)₄ is too short to accommodate the simultaneous binding of more than one macro domain.

2. What is the general PARylation state (albeit heterogeneous) of the nucleosome using the enzymatic method? Figure 1 suggests that they are either mono or di-ADP-ribosylated, but the cartoon in Figure 3 indicates longer and/or branched products.

Both histones H3 and H2B are PARylated by PARP1 and HPF1, with varying chain length and extent of branching (PMID 32939087), which results in a highly heterogeneous mixture of PARylated nucleosomes. For this reason and as outlined above, we consider it more prudent to omit the results previously obtained with enzymatically-PARylated nucleosomes. While all smFRET experiments demonstrated excellent reproducibility throughout, in stark contrast, the remodeling of enzymatically-PARylated nucleosomes displayed large variability between independent experiments with different nucleosome preparations. This became evident only as part of an overall effort to increase statistics for our smFRET analyses. We do not know the underlying reasons, and we can only surmise that the heterogeneity and complexity of the PARylation reaction might amplify subtle variations in conditions (such as, temperature or incubation time) to substantially impact the length, branching, and distribution of PAR chains. Identifying the sources of variation in this experiment would require considerable additional time and effort. We believe that the newly added experiments with exclusively homogeneously-PARylated histones allow for a much clearer analysis of remodeling directionality.

3. The example FRET trace in Figure 3b where the signal increased initially was followed by a decrease. Is this observed in other traces as well? If so, what is the explanation for it? The authors should show more example traces and perhaps perform more detailed analysis of the rich FRET data to provide more insights.

With DNA progressively moving into the nucleosome from the short linker side, the FRET efficiency is expected to initially increase but eventually reach maximum before decreasing again. For this reason, when observing an increase in FRET followed by a decrease (as in the example trace), we cannot be certain whether it arises from a reversal in remodeling direction after reaching high FRET values, or from processive movement towards the short linker. For remodeling away from the short-linker side, the signal is monotonous, however, such that a FRET decrease followed by a FRET increase represents a reversal in remodeling direction either within the same binding event, or with a new ALC1 molecule binding from solution. Indeed, we often observe such behavior, as evident from the new example traces we have now added (Figures 3b and S4a of the revised version of the manuscript). We now clarify this aspect with the following sentence in the results section:

“Since the FRET signal exhibits non-monotonic behavior in relation to nucleosome position when a nucleosome approaches the short DNA linker, we limit our analysis to the direction of the initial movement”.

Based on the reviewer’s comment, we have now additionally quantified the initiation time for remodeling under different conditions. Notably, the initiation time was approximately 3-fold longer when ALC1 was preincubated with nucleosomes as opposed to being added at the same time as ATP. Moreover, we observed the same trend with the closely-related Chd1 remodeler. This observation is unexpected, given that when the enzyme is added with ATP, the initiation might be limited by enzyme binding. One would expect shorter initiation times upon pre-incubation of the remodeler with nucleosomes. However, we observed the opposite effect, which may indicate that in the absence of ATP, both ALC1 and Chd1 bind to nucleosomes in an off-pathway conformation that persists for a relatively long time (tens of seconds) even after ATP is provided. We believe that this result is both mechanistically intriguing and of practical use for future research. We have therefore included the characterization of the remodeling initiation time in Figure S4b of the revised version of the manuscript, and added the following text to its results section:

“Interestingly, for both ALC1 and the closely related Chd1 remodeler, the remodeling initiation time was substantially shorter when the reaction was started by introducing the enzyme and ATP at the same time, compared to first allowing the enzyme to bind nucleosomes in the absence of ATP (Figure S4b). This suggests that in the absence of ATP, both ALC1 and Chd1 bind to nucleosomes in a long-lived off-pathway conformation”.

4. In Figure 3c, the distribution of FRET increase/decrease for the first column is expected to be somewhere between those for the second and third columns, because the ADP ribosylation on both sides of the nucleosome can induce sliding in opposite directions. However, the authors observed the highest fraction of FRET decrease of all conditions. This

could be due to the different lengths of the PAR chain as mentioned above, or PARylation of H3 as the authors alluded to. Please discuss this point.

We completely agree with the reviewer. Based on this comment, we have now generated nucleosomes with PAR chains on both copies of H2B and analyzed the directionality with which they are remodeled by ALC1. Notably, these symmetrically-ADP-ribosylated nucleosomes were shifted without any bias towards either side, making the corresponding data point lie in the middle between the data points for unique ADP-ribosylation on either side.

As outlined above, we have now removed the results previously obtained with enzymatically-PARylated nucleosomes, for which we observed a large variability between independent experiments with different nucleosome preparations. We believe that the newly added experiments with exclusively homogeneously-PARylated histones allow for a much clearer analysis of remodeling directionality.

We have now updated the text and Figures 3 and S4 of the revised version of the manuscript to incorporate these new results and the measures of statistical significance.

5. In the cryo-EM experiment, PARylation was installed on the longer-linker side, which seems counterintuitive because the paper argues that the shorter linker is the preferred side for PARP1/HPF1. Can the authors explain why this choice was made?

We thank the reviewer for spotting this mistake in the cartoon. Indeed, the PARylated H2B was installed on the short-linker side. We have now updated the figure accordingly and apologize for the oversight.

6. Line 127: "ALC1 is bound to the strong side (Figure 2c)..." Is this a typo (should be the "weak" side)?

We thank the reviewer for making us aware of this point of potential confusion. The weak-side H2B tail is closest to ALC1 when ALC1 is engaging the DNA at the strong side of the nucleosome. We have now updated the text to clarify this:

"Importantly, the local resolution in the DNA region is sufficient to distinguish purines from pyrimidines (Figure S2b), which enabled us to unambiguously determine that ALC1 is bound to the DNA at the strong side, close to the tail of the ADP-ribosylated H2B at the weak side (Figure 2c)".

7. The "short" and "long" linkers also have different meanings in different assays (Figure 1: 3/78; Figure 2: 0/10; Figure 3: 12/78). Is there a rule of thumb as to how short the linker needs to be in order to constitute a preferred site for ADP ribosylation?

We thank the reviewer for raising this important point. We have conclusively demonstrated that PARylation is preferentially initiated at the short linker-proximal H2B for the case when the short linker is three bp long. Repeating this assay for a range of linker lengths would be valuable for the field. Given the significant amount of time and effort required for this, we

believe that such experiments constitute an intriguing avenue for future studies. The cryo-EM and smFRET experiments were aimed at probing the consequences of asymmetric ADP-ribosylation. To this end, we generated asymmetrically-PARylated nucleosomes in a manner that does not rely on the linker length. Rather, the choice of the linker length for cryo-EM and smFRET analysis was dictated by other considerations. In the case of smFRET, a 12-bp linker ensured an optimal starting FRET value to provide sufficient dynamic range for the observation of nucleosome sliding in both directions.

Reviewer #3 (Remarks to the Author):

ALC1 is a single subunit chromatin remodeling enzyme that plays roles in DNA double strand break repair. ALC1 contains a Snf2-like ATPase domain, as well as a novel Macro domain that is known to bind to PAR chains that are attached to histones H3 or H2A, catalyzed by PARP enzymes. Previous studies from the authors' group and others, have shown that PARylation of histones enhances ALC1 activity by releasing the macro domain from an association with the ATPase lobes, and ALC1 prefers to slide an end-positioned nucleosome to the center (away from the ends). Furthermore, Deindl and colleagues published a 4 angstrom structure of ALC1 bound to a PARylated nucleosome (Bacic et al., 2021 ELIFE). This work also used a novel strategy to create nucleosomes where PARP enzymes preferred to modify histone sites adjacent to a short DNA end, mimicking the predicted modification of nucleosomes adjacent to a DSB (DNA contained a 5' phosphate on the short end, and a 5'-OH on the long end to control PARP activity). Ensemble FRET sliding assays with this substrate showed that PARylation could stimulate the ability of ALC1 to slide nucleosomes away from the DNA end.

In this current manuscript, Deindl and colleagues have extended their previous study by creating more homogeneous nucleosome substrates, combining an oriented hexosome assembly strategy and native ligation. In this way, they created nucleosomes where a single ADP modification was positioned on the histone H2A located adjacent to a short stretch of linker. Like their previous work, this substrate is predicted to mimic an in vivo location for PAR chains next to a DSB. Conversely they also made a substrate with the ADP on the opposite face of the nucleosome. Using an smFRET approach, the authors present evidence that the location of the PAR chain can influence the direction of nucleosome sliding by ALC1. Using one of these substrates, the authors also present a better cryoEM structure of ALC1 bound to a PARylated nucleosome (3 angstrom). But like previous structures, neither the PAR or the Macro domain of ALC1 are visualized.

In general, this work represents an incremental advance of our understanding of ALC1 action. The work confirms the major conclusions from their 2021 paper, perhaps with better reagents. This work reinforces the view that PARylation can direct ALC1 to bind to a preferred SHL2 position, and as expected this dictates the direction of sliding.

We thank the reviewer for their detailed comments. However, we respectfully disagree with their overall assessment of our manuscript, especially in relation to our previous paper (Bacic et. al. 2021, eLife, PMID 34486521). While it is true that the design of a cryo-EM substrate in our previous study was based on the hypothesis that PARP2 might preferentially PARylate histone tails closest to the active site, in the previous study we had no means of

probing whether this was indeed the case. Asymmetric PARylation was simply expected to produce a more homogenous sample, but was by no means necessary for a successful cryo-EM reconstruction. Similarly, the ensemble FRET-based nucleosome sliding assays as published in our previous paper invited us to speculate that asymmetric ADP-ribosylation could bias ALC1-induced nucleosome sliding. However, in the same paper, we were abundantly clear that we could not address this speculative hypothesis:

“Experimentally testing this hypothesis would require an asymmetric (H3-H4)₂ tetramer. In this tetramer, only a single copy of H3 would be fluorescently labeled, and the tetramer would exhibit a defined and known orientation on the DNA, with the labeled H3 proximal to either the long or short linker DNA. However, preparing such a nucleosome is currently not possible (reviewed in PMID 34319695). We note that we therefore cannot formally rule out alternative explanations, which do not involve asymmetric nucleosome PARylation, for the observed deviation from Michaelis–Menten kinetics of nucleosome sliding rates”.

We emphasize that rather than merely confirming the conclusions of our 2021 paper, we go much beyond our earlier findings. Using a completely new assay, we perform the, to the best of our knowledge, first analysis of the asymmetry in nucleosome ADP-ribosylation, presenting a nuanced and complex picture. Leveraging a new approach to generate asymmetrically, homogeneously, and site-specifically ADP-ribosylated nucleosomes, we mechanistically dissect the consequences of asymmetric ADP-ribosylation in cryo-EM and single-molecule FRET analyses. Our results suggest that the asymmetric introduction of ADP-ribose chains on the nucleosome could provide a means to simultaneously recruit and activate ALC1, and at the same time ensure preferential remodeling in the direction away from the closest DNA end. In our opinion, this current manuscript therefore represents a major step forward by directly addressing an important open question using new methodology that paves the way for future studies of nucleosome asymmetry.

Specific comments:

Figure 2. The figure should indicate where the long and short linkers are located with respect to the bound ALC1 and where the ADP is predicted to be located.

We thank the reviewer for this helpful suggestion. We have now updated Figure 2 with these labels.

Figure 3C. Here the authors need to show the data for the unmodified nucleosome substrate. They cite their older work, but that used a constitutively activated version of ALC1. The side-by-side comparison is essential here. The substrate is shown, the data should be available.

We completely agree with the reviewer and have now generated an H3(1-20)-(Ser10(ADPr)₄) peptide. Using this peptide, we were able to analyze the remodeling of unmodified nucleosomes by wild-type ALC1 activated in trans. We expected that under these conditions, wild-type ALC1 would remodel unmodified nucleosomes in the same manner as a constitutively-active ALC1 mutant that showed a preference for sliding nucleosomes away from the short linker in our previous experiments (PMID 33357431). Surprisingly, we observed an opposite preference when wild-type ALC1 was activated by ADP-ribose chains in trans. One potential explanation for these observations could be that

the macro domain can interact with linker DNA via its ADP-ribose binding site and act in a manner analogous to the DNA binding domain of Chd1 (PMID 21623345; PMID 33468676). However, in the presence of ADP-ribose chains engaging the ADP-ribose binding site, such an interaction with DNA is likely impossible, potentially resulting in an opposite remodeling directionality. Indeed, we have previously demonstrated that the macro domain of ALC1 can bind DNA (PMID 29220652). According to the current models of double-strand break repair, the break needs to be accessible to the repair machinery for successful processing and subsequent cell survival. Therefore, we believe that remodeling towards the DNA end is most likely detrimental for DNA repair, and this result emphasizes the importance of mechanisms for guiding ALC1 remodeling away from DNA ends. The asymmetric introduction of ADP-ribose chains on the nucleosome could provide a means to simultaneously recruit and activate ALC1, and at the same time ensure preferential remodeling in the direction away from the closest DNA end. We have now updated Figures 3 and S4, and the manuscript text to incorporate the new results.

Figure 3. In previous work, the authors showed by ensemble FRET studies that an asymmetrically modified nucleosome yielded biphasic remodeling kinetics, as a function of ALC1 concentrations. At low concentration, faster rates were found, then higher concentrations revealed a slower rate of sliding. This was interpreted as ALC1 having different binding affinities for the two SHL2 locations – one location had higher affinity due to the presence of the PAR group on H3/H2A. Similar studies should be shown here. The expectation is that ALC1 will also occupy the SHL2 position that is not directed by ADP. The difference between the SHL2 occupancies is key, as the cryoEM conditions were established with an excess of ALC1 per nucleosome.

We thank the reviewer for this suggestion and agree that this would be useful. Unfortunately, however, such ensemble titration experiments would require impractically large quantities of ADP-ribosylated histones.

The question of whether multiple ALC1 molecules can bind to a nucleosome containing a single ADP-ribose chain can be directly addressed using native PAGE. The native gel for the ALC1 complex with an enzymatically-ADP-ribosylated nucleosome that was subjected to cryo-EM analysis in our previous study shows multiple bands, indicating heterogeneous complex formation. In stark contrast, the corresponding gel for the ALC1 complex with a defined, asymmetrically, and site-specifically ADP-ribosylated nucleosome exhibits a single band (Figure R1). This is consistent with the much-increased homogeneity of the cryo-EM data presented in the current manuscript. Additionally, we observed essentially no nucleosome sliding in smFRET experiments with unmodified nucleosomes and wild-type ALC1 in the absence of the ADP-ribosylated peptide.

Furthermore, we have now used the recently published algorithm OccuPy (PMID 37726277) to quantify the occupancy of the features of our cryo-EM map. This new analysis revealed that the acidic patch on the entry side contains additional density, while the acidic patch on the other side does not. This result further corroborates that most likely only one ALC1 molecule can bind a nucleosome with a unique tri-ADP-ribose chain.

Taken together, these data strongly suggest that wild-type ALC1 is unable to bind nucleosomes without PAR chains either on the nucleosomes themselves or added in trans,

consistent with the auto-inhibition mechanism we and others characterized previously (PMID 29220652; PMID 29220653).

In our earlier manuscript we assumed the simplest scenario, where only one side of the nucleosome is ADP-ribosylated. We stress that in this earlier paper, we were exceedingly clear about the fact that we had no direct evidence for the extent of PARylation on each side and that our model was only based on indirect evidence. However, our direct analysis of asymmetric PARylation now reveals a more nuanced picture. In light of our new data, the likely explanation for the previously-observed biphasic behavior is that the two sides of the nucleosomes are ADP-ribosylated to different extents, resulting in different ALC1 affinities towards the two SHL2 sites.

Figure R1. Native PAGE analysis for complexes between ALC1 and enzymatically-ADP-ribosylated nucleosomes (left) or site-specifically ADP-ribosylated nucleosomes (right). Lane 1: marker, lane 2: nucleosomes, lane 3: nucleosomes and ALC1.

Intro, line 65 and 68. These two sentences seem incompatible. Are the authors referring to the 4 angstrom structure as not “high-resolution”? Perhaps this could be expanded. The current “high-resolution” structure has allowed the visualization of the DNA bases, but not much else.

We agree with the reviewer that our new cryo-EM structure is sufficiently improved in resolution to allow us to infer the orientation of the 601 sequence. This is crucial, since unlike in our earlier paper, in this current study we are structurally examining the consequence of asymmetric ADP-ribosylation using a nucleosome substrate of defined asymmetry. Therefore, while clearly improved compared to our previous work, the absolute, nominal resolution of the cryo-EM reconstruction is not particularly relevant for the mechanistic

analysis in this paper. In light of the reviewer's comment, we have therefore removed from the introduction section the sentence, "However, there is, as of yet, no high-resolution structure of ALC1 bound to an ADP-ribosylated nucleosome."

In their previous work, the authors stated "The structure of PARP2 and HPF1 bound to nucleosomes indicates that, of the two H3 tails in a nucleosome, the one on the side of the DNA end bound by PARP2 is closest to its active site (Bilokapic et al., 2020). Target residues for ADP-ribosylation (Ser in KS motifs) in this proximal H3 tail should therefore be favored over the equivalent residues in the distal H3 tail." Doesn't this predict the data shown in Figure 1? Perhaps this should be mentioned here as well?

We thank the reviewer for pointing this out. Indeed, the data presented in Figure 1 directly probe a hypothesis that we put forward in our previous paper, albeit in relation to H2B PARylation rather than H3, presenting a nuanced and complex picture of nucleosome ADP-ribosylation. We have now added to the discussion section a reference to the Bilokapic et al. 2020 structure to the discussion:

"This result is consistent with a structure of PARP2 and HPF1 bound to a nucleosome (PMID 32939087), which suggests that the H3 and H2B tails closest to the active site are likely to be favored for modification".

Based on the cryoEM data, the authors conclude that the location of the ADP-H2A adduct dictates which SHL2 is bound by ALC1. However, in the absence of a nucleosomal substrate with the label on the opposite face, it remains a possibility that the short linker is the determinate, not ADP.

We thank the reviewer for bringing up this important point. Indeed, we cannot formally rule out that the short linker could also contribute to the binding preference of ALC1. However, we consider such a scenario unlikely, based on the following considerations:

- 1) Since the macro domain is engaged with an ADP-ribose chain, the motor is located at SHL2, and the ALC1 linker engages the acidic patch, it is difficult to imagine what part of the ALC1 protein could interact with the DNA linker, for the DNA linker length to have such an effect.*
- 2) When ALC1 is activated by a PARylated peptide in trans, it has a preference for sliding nucleosomes towards rather than away from the shorter linker, which suggests that in the absence of nucleosome ADP-ribosylation, ALC1 has a preference for a binding orientation that is opposite to that observed in our cryo-EM structure.*

In light of the reviewer's comment, we have now added the following cautionary note to the discussion section of the manuscript:

"We cannot formally rule out the possibility that the DNA linker length could contribute to such a binding preference. However, we consider such a scenario unlikely, since when ALC1 is activated by a PARylated peptide in trans, it has a preference for sliding nucleosomes towards rather than away from the shorter linker. This suggests that in the absence of

nucleosome ADP-ribosylation, ALC1 has a preference for a binding orientation that is opposite to that observed in our cryo-EM structure”.

In the smFRET studies, traces where the nucleosome is moved towards the edge (increased FRET) shows only a transient increase, followed by decreased FRET. This would imply that ALC1 rapidly re-orient to the other SHL2? Is it still bound to PAR? Are the rates for this subsequent decrease slower than the initial movement? If the macro domain releases PAR when bound to the other SHL2, one would assume inhibition?

With DNA progressively moving into the nucleosome from the short linker side, the FRET efficiency is expected to initially increase but eventually reach maximum before decreasing again. For this reason, when observing an increase in FRET followed by a decrease (as in the example trace), we cannot be certain whether it arises from a reversal in remodeling direction after reaching high FRET values, or from processive movement towards the short linker. For remodeling away from the short-linker side, the signal is monotonous, however, such that a FRET decrease followed by a FRET increase represents a reversal in remodeling direction either within the same binding event, or with a new ALC1 molecule binding from solution. Indeed, we often observe such behavior, as evident from the new example traces we have now added (Figures 3b and S4a of the revised version of the manuscript). We now clarify this aspect with the following sentence in the results section:

“Since the FRET signal exhibits non-monotonic behavior in relation to nucleosome position when a nucleosome approaches the short DNA linker, we limit our analysis to the direction of the initial movement”.

We observe remodeling in both directions within a single trace with nucleosomes that possess only one PAR chain, and no remodeling without any PAR chains present. Most likely, PAR chains on one side allow ALC1 to engage both SHL2 locations, albeit with different probabilities. Further characterization of the ability of ALC1 to switch remodeling directions would require considerable additional time and effort, and could, in our opinion, form the basis of interesting future studies.

Reviewer #1 (Remarks to the Author):

The revised manuscript is improved, with nicely presented experimental findings in Figures 1 and 2. However, Figure 3 does not seem to give a clear cut answer, which is important because it is the main conclusions of the paper.

The authors use the results from the smFRET experiment in Fig 3 to state that ALC1 has a preference for shifting nucleosomes directionally by acting to the PARylated side. Unfortunately, the effect is not large (57% away from the ends for the one with PAR on the short end dimer, compared with 51% away from the ends when no PAR labeling). Looking at the traces provided, the alternation between high and low FRET states would seem to suggest that the nucleosomes are shifted in both directions by ALC1, despite asymmetric labeling. Perhaps there is a biochemical difference for these PAR-labeled nucleosomes that is less apparent with a smFRET experiment? I assume that the authors see no significant differences in migration on native gels, or overall rate of sliding. If the authors haven't already done so, it might be useful to see if PAR-labeled nucleosomes are preferentially shifted (faster kinetics) when mixed with nonPAR nucleosomes.

Figure 3 shows two example traces, and it seems odd that in one, the FRET values shift from the starting point (~ 0.5) to higher FRET, back and forth, whereas in the other, FRET values shift from the starting point to lower FRET, but also back and forth. Is that consistently seen or just random for these two examples?

The authors present a strong and attractive model, however, the results seem to be suggesting that there is more to the story, probably something more subtle or nuanced.

minor comment: the positions of the H2B tails in the Fig 3a cartoon (right half) seem to be swapped – orange should be on left and yellow on right.

Reviewer #2 (Remarks to the Author):

The authors have fully addressed my concerns. The usage of homogeneously PARylated nucleosomes clarified a lot of the confusions and made the manuscript stronger. I now recommend its publication.

Reviewer #3 (Remarks to the Author):

The appreciate the thoughtful and honest response of the authors to my previous comments. Changes to the text, and the addition of several new experiments greatly strengthen this work. I am now highly enthusiastic towards the publication of this work in Nature Communications. This piece will have a major impact on the chromatin remodeling community, as well as providing new insights to the regulation of DNA double strand break repair.

Detailed point-by-point response to the reviewers' comments

Reviewer #1 (Remarks to the Author):

The revised manuscript is improved, with nicely presented experimental findings in Figures 1 and 2. However, Figure 3 does not seem to give a clear cut answer, which is important because it is the main conclusions of the paper. The authors use the results from the smFRET experiment in Fig 3 to state that ALC1 has a preference for shifting nucleosomes directionally by acting to the PARylated side. Unfortunately, the effect is not large (57% away from the ends for the one with PAR on the short end dimer, compared with 51% away from the ends when no PAR labeling).

We thank the reviewer for their comments. Regarding Figure 3, the value of 51% actually refers to the doubly-PARylated nucleosome and indicates the lack of any directional preference in the presence of symmetric PARylation. In the absence of nucleosome PARylation, 39% (rather than 51%) of the traces display an initial movement away from the short linker DNA. In stark contrast, with asymmetric nucleosome PARylation, the fraction of such traces is 57%. Importantly, asymmetric PARylation therefore reverses the directional bias: in the absence of histone PARylation, the nucleosome is preferentially shifted towards its short-linker side, whereas asymmetric PARylation leads to an overall movement away from that DNA end. The effect is therefore not only highly statistically significant, but also results in a switch in the overall remodeling direction, which is sufficient to create an initial net movement away from the break site. We apologize for not having made this sufficiently clear.

Based on the reviewer's comment, we have now clarified this with the following statement in the results section of the revised manuscript: "Indeed, among nucleosomes featuring a single, site-specifically and homogeneously ADP-ribosylated dimer on the shorter linker-proximal side, a majority (57%) of individual traces featured an initial decrease in FRET. This is consistent with ALC1-induced sliding of nucleosomes away from the shorter linker and in stark contrast to just 39% of such traces when PAR chains were provided in trans (Figure 3c)."

Looking at the traces provided, the alternation between high and low FRET states would seem to suggest that the nucleosomes are shifted in both directions by ALC1, despite asymmetric labeling.

The FRET time traces of nucleosome remodeling often exhibit non-monotonic behaviors beyond the initial monotonic change. However, with DNA progressively moving into the nucleosome from the short linker side, the FRET efficiency is expected to initially increase but eventually reach maximum before decreasing again. For this reason, when observing an increase in FRET followed by a decrease, we cannot be certain whether it arises from a reversal in remodeling direction after reaching high FRET values, or from processive movement towards the short linker. For remodeling away from the short-linker side, the signal is monotonous, such that a FRET decrease followed by a FRET increase represents a reversal in remodeling direction either within the same binding event, or with a new ALC1

molecule binding from solution. Indeed, we often observe such direction reversals for all nucleosomes, including the ones with a single PAR chain. Due to the above-mentioned ambiguity of the FRET signal, we have limited our analyses to the initial remodeling direction, which is sufficient for probing a bias in remodeling directionality. Indeed, our data indicate that the majority (57%) of nucleosomes with a single PAR chain on the short linker-proximal side are initially remodeled away from the short linker, while a significant fraction (43%) are initially remodelled towards it. Most likely, a single PAR chain on H2B allows ALC1 to engage the nucleosome at both SHL2 locations, albeit with different probabilities. We note that the observed reversals in remodeling directionality are fully consistent with these results. It is intriguing to speculate that the ability to frequently switch remodeling directionality could help ensure that all nucleosomes are remodeled away from the DNA break. In the opposite scenario, where ALC1 remodeling were committed to a single direction for a long period of time, it would be critical (albeit likely impossible) to ensure a perfect accuracy in always engaging the correct SHL2 location. Otherwise, a fraction of nucleosomes shifted towards a DNA break would remain there for extended periods of time, which would likely impede the efficient repair of the DNA break.

Perhaps there is a biochemical difference for these PAR-labeled nucleosomes that is less apparent with a smFRET experiment? I assume that the authors see no significant differences in migration on native gels, or overall rate of sliding.

All PARylated nucleosomes used for the smFRET analyses in this study contain the same DNA sequence, fluorophores, histones, and the same synthetic PAR chain of precisely defined length, with the only difference being the location of the PAR chain(s). By design, there is therefore no biochemical difference between these nucleosomes aside from the location of the PAR chain. Nonetheless, in light of the reviewer's comment, we have now carried out additional analyses. The FRET histograms of all these nucleosomes are virtually identical with a single FRET peak at around 0.5, indicating the absence of significant structural differences among the different nucleosomes (Fig. R2). Moreover, the fluorescence intensities from donor and acceptor dyes as well as their photobleaching behavior show that homogenous populations of mononucleosomes were observed, without any differences in oligomeric status among the different nucleosomes.

Figure R2. Starting FRET histograms for smFRET nucleosomes without PAR chains, with unique PAR chains on the entry- or exit-side H2B, or both.

If the authors haven't already done so, it might be useful to see if PAR-labeled nucleosomes are preferentially shifted (faster kinetics) when mixed with nonPAR nucleosomes.

We have tested the ability of ALC1 to mobilize non-PARylated nucleosomes and observed no remodeling in the absence of PAR chains. The only scenario where we could envision ALC1-induced sliding of non-PARylated nucleosomes in the presence of PARylated nucleosomes would involve the activation of ALC1 by PAR chains in trans. However, we consider such a scenario highly unlikely, based on the following rationale. Via its macro domain, ALC1 would be tethered to the nucleosomal PAR chains, resulting in an extremely high local concentration of SHL2 sites from the PARylated nucleosomes. Consequently, non-PARylated nucleosomes would also have to be present at extremely high concentrations for them to be able to compete off and displace the ALC1 ATPase from the PARylated nucleosome.

We therefore consider it safe to assume that PARylated nucleosomes would represent the by-far-preferred substrate for ALC1-induced remodeling, even in the presence of non-PARylated nucleosomes.

Figure 3 shows two example traces, and it seems odd that in one, the FRET values shift from the starting point (~0.5) to higher FRET, back and forth, whereas in the other, FRET values shift from the starting point to lower FRET, but also back and forth. Is that consistently seen or just random for these two examples?

The example traces presented in the figures are representative of our data. Indeed, we frequently observe changes in the remodeling directionality, although the nature of the FRET signal hampers the interpretation of the non-monotonous FRET changes. For a detailed description of this aspect, please see our answer above.

The authors present a strong and attractive model, however, the results seem to be suggesting that there is more to the story, probably something more subtle or nuanced.

The repair of DNA breaks is a very complex process, and many other layers of regulation and additional players are likely involved. While we therefore fully agree with the reviewer that likely 'there is more to the story', we believe that we have conclusively investigated and demonstrated an important aspect of how ALC1 could contribute to clearing DNA break sites. We eagerly anticipate future studies.

minor comment: the positions of the H2B tails in the Fig 3a cartoon (right half) seem to be swapped – orange should be on left and yellow on right.

We thank the reviewer for noticing this mistake and apologize for the oversight. We have now corrected the colors of the H2B tails in Fig. 3a.

Reviewer #2 (Remarks to the Author):

The authors have fully addressed my concerns. The usage of homogeneously PARylated nucleosomes clarified a lot of the confusions and made the manuscript stronger. I now recommend its publication.

We thank the reviewer again for their positive evaluation of our manuscript and for their very insightful and constructive suggestions.

Reviewer #3 (Remarks to the Author):

The appreciate the thoughtful and honest response of the authors to my previous comments. Changes to the text, and the addition of several new experiments greatly strengthen this work. I am now highly enthusiastic towards the publication of this work in Nature Communications. This piece will have a major impact on the chromatin remodeling community, as well as providing new insights to the regulation of DNA double strand break repair.

We thank the reviewer for their very insightful and constructive suggestions and for their enthusiasm and positive evaluation of our manuscript.

Reviewer #1 (Remarks to the Author):

I appreciate the sincere efforts by the authors to explain and improve the manuscript. The authors' rebuttal has helped me understand several points, and I agree that the manuscript is a significant advance that is worthy of publication in Nature Communications. I make some suggestions below, but would not be opposed to publication in its present form.

One point that remains unclear is why the ADPr in trans showed a distinct effect from the dual modified nucleosomes (in terms of directionality). It seems to suggest some constraint when the modification is on the histones, which may be worth pointing out. My main concern was that the single-labeled histones should be compared with the double labeled nucleosome, not the nucleosome with ADPr in trans; and for that comparison of single-ADPr to double ADPr, the change in initial sliding direction is much more modest. The authors might consider removing the language "stark contrast" to emphasize this point with the trans ADPr experiment, as it may not be the most appropriate comparison to be making here.

The authors might consider including in the Discussion a mention of how dependent ALC1 is on ADPr modifications, and that a single mark on one side that allows sliding in both directions means that the remodeler can act on both sides. It may still be true that the bias in sliding direction may not be the major purpose of the PAR chains, that they mainly direct ALC1 where to go. It may therefore be helpful to fold the current story of direction bias into that context.

Minor point – the coloring for Fig 3 is now correct, but the ADPr labels left over from the previous version are now incorrect; in particular, the third cartoon on the right that states labeling on the short linker, it shows ADPr on an orange tail, whereas the labeling should be on the yellow one.

Detailed point-by-point response to Reviewer #1's comments

Reviewer #1 (Remarks to the Author):

I appreciate the sincere efforts by the authors to explain and improve the manuscript. The authors' rebuttal has helped me understand several points, and I agree that the manuscript is a significant advance that is worthy of publication in Nature Communications. I make some suggestions below, but would not be opposed to publication in its present form.

We thank this reviewer again for their thorough review and insightful comments, which we believe have substantially improved our manuscript.

One point that remains unclear is why the ADPr in trans showed a distinct effect from the dual modified nucleosomes (in terms of directionality). It seems to suggest some constraint when the modification is on the histones, which may be worth pointing out.

We agree with the reviewer. The macro domain movement is likely substantially more constrained when the PAR chains are located on histones as compared to when the PAR chains are provided in trans. This constraint is most likely key to explaining the difference in remodeling directionality between the symmetrically-PARylated nucleosomes and the scenario where PAR chains are added in trans. We have now added the following sentence to the Discussion:

"It is notable that ALC1 has a preference for sliding nucleosomes towards the short linker when PAR chains are added in trans, but has essentially no preference when sliding symmetrically-PARylated nucleosomes. One of the main differences between these two cases is the fact that the macro domain movement is likely substantially constrained when PAR chains are attached to histones. These results suggest an intriguing possibility: that the directionality of ALC1 remodeling can be influenced by the reach of the macro domain."

My main concern was that the single-labeled histones should be compared with the double labeled nucleosome, not the nucleosome with ADPr in trans; and for that comparison of single-ADPr to double ADPr, the change in initial sliding direction is much more modest. The authors might consider removing the language "stark contrast" to emphasize this point with the trans ADPr experiment, as it may not be the most appropriate comparison to be making here.

In light of the reviewer's comment we have now rephrased the respective part of the Results section:

"This is consistent with ALC1-induced sliding of nucleosomes away from the shorter linker, which notably differs from only 39% of such instances observed when PAR chains were provided in trans (Figure 3c). Conversely, nucleosomes that featured the site-specific ADP-ribosylation on the longer linker-proximal side were preferentially shifted towards the shorter linker (46% of traces with initial FRET decrease versus 57% for the shorter-linker proximal side)."

The authors might consider including in the Discussion a mention of how dependent ALC1 is on ADPr modifications, and that a single mark on one side that allows sliding in both

directions means that the remodeler can act on both sides. It may still be true that the bias in sliding direction may not be the major purpose of the PAR chains, that they mainly direct ALC1 where to go. It may therefore be helpful to fold the current story of direction bias into that context.

We appreciate these suggestions. We have now updated the discussion section to highlight the fact that ALC1 can remodel nucleosomes with a single PAR chain in both directions: “Interestingly, even a single minimal PAR chain enables ALC1 remodeling in both directions, albeit with different probabilities. Most likely, the enzyme can switch remodeling direction without completely dissociating from the nucleosome.”

We also emphasize the critical role of histone PARylation in recruiting and activating ALC1: “Besides playing a critical role in promoting the efficient recruitment of ALC1 to and activation at lesion-proximal nucleosomes, histone PARylation now also appears to help ensure the required direction of sliding.”

Minor point – the coloring for Fig 3 is now correct, but the ADPr labels left over from the previous version are now incorrect; in particular, the third cartoon on the right that states labeling on the short linker, it shows ADPr on an orange tail, whereas the labeling should be on the yellow one.

We thank the reviewer for noticing this mistake and apologize for this additional oversight. We have now updated the figure to correct it.